# Out-of-Distribution Detection via Conditional Kernel Independence Model

**Yu Wang**[*1] **, Jingjing Zou**[*2]**, Jingyang Lin**[3]**, Qing Ling**[3]**, Yingwei Pan**[4]**, Ting Yao**[4]**, Tao Mei**[4]

[1]: Qiyuan Lab, Beijing, China
[2]: University of California, San Diego, USA
[3]: Sun Yat-sen University, Guangzhou, China
[4]: JD AI Research, Beijing, China

`feather1014@gmail.com, j2zou@ucsd.edu, yung.linjy@gmail.com`
`lingqing556@mail.sysu.edu.cn, {panyw.ustc, tingyao.ustc}@gmail.com, tmei@jd.com`

## Abstract

Recently, various methods have been introduced to address the OOD detection problem with training outlier exposure. These methods usually count on discriminative softmax metric or energy method to screen OOD samples. In this paper, we probe an alternative hypothesis on OOD detection by constructing a novel latent variable model based on independent component analysis (ICA) techniques. This novel method named Conditional-i builds upon the probabilistic formulation, and applies the Hilbert-Schmidt Independence Criteria that offers a convenient solution for optimizing variable dependencies. Conditional-i exclusively encodes the useful class condition into the probabilistic model, which provides the desired convenience in delivering theoretical support for the OOD detection task. To facilitate the implementation of the Conditional-i model, we construct unique memory bank architectures that allow for convenient end-to-end training within a tractable budget. Empirical results demonstrate an evident performance boost on benchmarks against SOTA methods. We also provide valuable theoretical justifications that our training strategy is guaranteed to bound the error in the context of OOD detection. Code is available at: https://github.com/OODHSIC/conditional-i.

## 1 Introduction

The out-of-distribution (OOD) detection ability of models is at the core of achieving machine learning safety and reliability. Knowing how the classifiers perform on unseen OOD data remains as a challenging task, owing to the train-test distribution divergence [3]. Conventional computer vision tasks such as [8, 24, 36, 47, 53, 54, 55, 64, 65, 66, 70, 77], NLP tasks [15, 74, 75] and other machine learning tasks therefore need to carefully take into account the effect of potential distribution shift between training and test data. While deep generative models have become immensely popular, seminal OOD detection works consider applying generative modeling techniques to reconstruct the training data distribution density, so that statistical test can be implemented on the density. However, it is found that such inference can perform strong prediction artifact owing to various possible reasons [12, 48, 50, 67, 78]. Instead, another line of work considers deploying discriminative methods on OOD detection [28, 34, 37, 38, 41] where the detection is implemented by thresholding some class uncertainty measurements obtained from training inlier classifiers. However, as the train and test data i.i.d. assumption is severely violated for the OOD detection task, only limited theoretical guarantees hold when deploying these discriminative models. There has been a surge of interest in the OOD detection task recently [9, 18, 40, 49, 56, 58, 59, 60, 69].

---

[*]Yu Wang and Jingjing Zou contributed equally to this work (joint first author).

36th Conference on Neural Information Processing Systems (NeurIPS 2022).

Unlike previous efforts that rely on canonical discrimination or generative models, we view the OOD detection problem from a new perspective of the independent component analysis (ICA). Our starting point is the hypothesis that OOD data should exhibit a slight dependency on in-distribution training data, which can effectively constrain the OOD data to expose limited mutual information with the inliers. This hypothesis imposes the inliers to extract only little predictive information from outliers during training, and we distinguish OOD samples via the related independence measurements during the test. The contemporary work [39] attempted to probe the OOD problem from a similar view. However, [39] merely demonstrates the promising practical values of such independence assumption based on empirical success, which lacks theoretical justification and supporting analysis. In this work, we propose a brand new motivating generative model, by incorporating additional class conditions into the latent variable model. This new work also aims to showcase the theoretical soundness through the transparent lens of ICA techniques, which provides desired technical convenience.

Our contribution to this paper includes: C1: We propose a new OOD detection framework called Conditional-i model (reminiscent of **Conditional-i**ndependence model). In comparison to [39], Conditional-i additionally encodes the class condition into the probabilistic dependence model. C2: In order to exclusively facilitate the end-to-end training of the Conditional-i model, we construct an efficient memory bank architecture that constrains the training within a practical computational budget. C3: Conditional-i can be efficiently implemented both when training OOD data is available or not. C4: We provide theoretical justifications for our new model. Empirical results demonstrate the evident superiority of our new proposal over the state-of-the-art methods for computer vision and NLP tasks.

## 2 Related Work

**Challenges for OOD detection**. In [12, 28, 50, 67], empirical evidence shows that deep generative models trained on image datasets can often erroneously assign a high likelihood to OOD inputs. In the meanwhile, as observed in [23, 33], discriminative models also encounter a similar issue in returning high confidence on OOD samples or misclassified samples. Many contemporary works attempted to understand the principle behind this phenomenon. The work in [52] figured that the shared background statistics between in-distribution data and OOD data could interfere with the test. In [50, 67], they discussed the impact of *typicality* on the the test. In [78], the work rather challenged the prevailing typical set hypothesis and the assumption that in-and out-distribution overlap. In [76], authors argued both accurate density estimation and discriminative classifier are critical, and henceforth proposed a hybrid model to address the issue. Research works also aim to alleviate the issue from the view of miscalibration [16, 19, 23, 44]. The work in [18] further demonstrated that deep architecture on the OOD detection performance task could have a significant impact.

**OOD detection with statistical uncertainty**. Adapting NNs to incorporate uncertainty and probabilistic methods has been of great interest in the community. The majority of this line of work rely on a Bayesian formalism, called Bayesian Neural Net (BNN). Many extensions along this path aim to estimate a posterior distribution over the DNN parameters [5, 20, 42] given some prior distribution on the DNN parameters. However, Bayesian NNs are often harder to implement and are prohibitively expensive in computational complexity. Some approach then considers reflecting a similar spirit of BNN [33] with reduced complexity. In the meanwhile, the statistical tests in differentiating OOD data are also of critical importance. In [1, 57], hierarchical probabilistic models are used to estimate uncertainty on OOD data. In [38], authors found that in-distribution data are more sensitive to softmax temperature scaling and noise perturbations, under which they proposed a novel statistical test. In [34], the Mahalanobis distance is estimated to reflect the uncertainty of the test data classes. There are many more explorations along this path [26, 27, 29, 38, 40, 41, 57]. Particularly, a group of works assumes some OOD samples are accessible during training [39, 26, 28, 37, 41]. The approach in [28] assigned the "uniform label" on OOD training data to impose the uncertainty on OOD data. The model in [41] introduced an energy-based score to replace the softmax score, achieving superior performance. The work in [37] investigated sampling policies of the OOD training data to improve OOD detection performance.

Conditional-i is the first that considers the independence assumption to address the OOD detection problem. Measuring independence and non-linear relationship between features is a classic technique, while independence component analysis (ICA) has powered conventional machine learning techniques [35, 43, 63]. Some emerging work, such as in [79] empirically shows removing non-linear correlation

between features improves generalization for deep models. However, these models were built and motivated for drastically different tasks. For instance, [79] targets the specific domain generalization (DG) task, whereas Conditional-i is tailored for the OOD detection task, two entirely different tasks having distinct definitions, modeling assumptions, and evaluation metrics. In comparison to our earlier contemporary work HOOD [39], this paper is further motivated by a modified conditional independence latent variable model. The fresh generative model behind Conditional-i provides unique technical convenience in delivering theoretical guarantees by allowing Conditional-i to utilize more flexible outlier regularizers during training.

## 3 Methodology

We introduce variable $x \in \mathbb{R}^D$ to denote in-distribution (inlier) data, and let variable $u \in \mathbb{R}^D$ represent the out-of-distribution (outlier) data. The in-distribution training data set $\mathcal{D}_{in}^{tr}$ consists of annotated pairs of data $(x_i, y_i^{in})$, where $i$ indexes training samples. The label $y_i^{in} \in \{1, ..., C\} = \mathcal{Y}_{in}$ assigns each $x_i$ one out of the $C$ ground truth classes. A common assumption for OOD detection is that some out-of-distribution (OOD) samples $u$ can be accessed during training, where $u$ are defined as samples from distinct classes $y_i^{out}$ [28, 41]. We cannot access labels of outlier $u_i$ samples.

We define $z_i = f(x_i; \theta) \in \mathbb{R}^d$ to be the features of inliers $x_i$. Function $f(\cdot; \theta)$ is the deep feature extractor with deep parameters $\theta$ to be learned. The features of OOD sample $u_i$, denoted as $q_i$ are also learned through the same feature extractor, i.e., $q_i = f(u_i; \theta) \in \mathbb{R}^d$. Out-of-distribution detection is essentially a binary classification task that aims to learn efficient $f(\cdot; \theta)$ that can separate and discriminate between inliers $z$ and outliers $q$ at test time.

### 3.1 Motivating principle

In this paper, OOD samples are defined to be from the classes outside the inlier classes $\mathcal{Y}_{in}$, by strictly following discriminative models [28, 34, 38, 41]. This definition is distinct from the generative model based detection methods [12, 50, 67, 78] that samples from a different feature distribution are instead defined to be OOD data regardless of their classes. Mainstream discriminative OOD detection approaches usually rely on softmax metric [28] or the likes, e.g., energy score [41] to make decisions. These methods essentially compare the linear correlation between a feature and the classifier weights.

Our aim is to reduce the mutual information between training inliers and training outliers, since features representing different semantics should confine their predictive ability to each other. Note that reducing mutual information between variables implies reduced dependence, while dependence captures a more general and complicated statistical relationship than linear correlation. As a partial step towards this goal, our new model relies on an explicit conditional independence hypothesis that is shown to be both theoretical sound and practically superior to the SOTA methods.

Pushing further than our contemporary work formulated as "HOOD" [39], our new OOD detection training strategy is built on a novel generative model under particular conditional independence assumption. Specifically, we define each sample from the $c$ class to be modeled as a variable $x^{(c)}$ generated from a generative process $g^{(c)} : \mathbb{R}^d \to \mathbb{R}^D$, $x_i^{(c)}$ is the $i^{th}$ sample realization of variable $x^{(c)}$:

$$x_i^{(c)} = g^{(c)}(w_i^{(c)}), \quad u_i = g^{(o)}(\beta_i). \quad (1)$$

Here, $w_i^{(c)}$ are i.i.d. realizations of variable $w^{(c)}$, and $w^{(c)}$ defines the classifier that reflects the common components shared between each sample in $c$ class. Variable $\beta_i$ denotes the latent variable defining the generative process of the outliers $u$ through function $g^{(o)}$. Each $\beta_i$ is i.i.d. sample with finite first and second moments. This suggests that we could model $z$ across all classes $c \in \{1, ..., C\}$, each denoted as $z^{(c)}$, along with $q$ variable to be jointly distributed given $\theta$:

$$(z^{(1)}, ..., z^{(c)}, ..., z^{(C)}, q) \sim p(z^{(1)}, ..., z^{(c)}, ..., z^{(C)}, q; \theta). \quad (2)$$

We further define variable $\xi$ as the variable indicating the true class of feature $z$. The relevance of this construction then becomes apparent as per the following objective on requiring the independence between $z$ and $q$ given class $c$:

$$\mathcal{R}(z, q, c, \theta) = \text{MMD} \, [\, p_\theta(z^{(\xi)}, q | \xi = c) \, || \, p_\theta(z^{(\xi)} | \xi = c) \cdot p_\theta(q) \,], \forall c, \quad (3)$$

where $p_\theta$ denotes a probability density parameterized by $\theta$. Penalizing maximum mean dependency (MMD) [21] between the joint distribution on $z^{(\xi)}, q$ and product of two conditional distributions

in Eq. (3) then explicitly regularizes the training so that inlier feature $z$ of every class $\xi = c$, to be independent with outlier feature $q$. Deep parameter $\theta$ is trained so that unseen OOD test data can generalize conditional independence given $\xi = c$, and only lives in the constrained subspace subject to small loss incurred from Eq. (3). Contradiction against independence assumptions for any class $\xi = c$ would increase dependence metric between $q$ and the generative process defined by $\xi = c$.

The model above ultimately allows us to incorporate the class condition and the associated generative model into an integrated loss function, one that can be directly exploited and practically penalized. It is of pivotal importance to note that Eq.(3) strongly contrasts with the sampling strategy in HOOD [39]. It also manifests that since $p_\theta(z, q) = \sum_c p_\theta(z^{(\xi)}, q | \xi = c) p(\xi = c)$, then sampling from $p_\theta(z^{(\xi)}, q | \xi = c)$ here by conditioning on class $\xi$ further eliminates the sampling artifact introduced by $p(\xi = c)$ of the training dataset. In this regard, conditioning on each specific $c$ avoids implicit marginalization over $p(\xi = c)$ as in [39] and instead encourages all in-distribution classes to produce independent features from outlier classes. Another compelling reason is that, such additional conditioning also provides technical convenience in delivering our theoretical guarantees by linking to the generative model of each specific class, see Theorem 1.

### 3.2 Conditional independence model for OOD detection

As proposed in [22], the Hilbert-Schmidt Independence Criterion (HSIC) is an effective measurement that quantifies the variable dependence through the eigenspectrum of covariance operators in reproducing kernel Hilbert spaces (RKHSs). In [22], it is guaranteed that the HSIC metric is zero if and only if the random variables are independent. We therefore anchor our modeling to this principled criteria, and employ the HSIC metric to encourage the conditional independence between $z$ and $q$ given each observed class $c$ of $z$. To reach this goal, during each training iteration, we sample pairs of samples $\mathcal{S}_{c,N} = \{(z_1^{(c)}, q_1), (z_2^{(c)}, q_2), ..., (z_i^{(c)}, q_i), ..., (z_N^{(c)}, q_N)\}$ from each conditional distribution $p_\theta(z^{(\xi)}, q | \xi = c)$ given each class $c$ and current $\theta$. Here, inlier samples $z_1^{(c)}, ..., z_N^{(c)}$ out of set $\mathcal{S}_{c,N}$ are from the same ground truth class $c$. Then, for each class $c$, Eq. (3) can be empirically estimated via the estimator modified from [22]:

$$\hat{\mathcal{R}}(z, q, c, \theta) = \frac{1}{(N-1)^2} tr(\Phi_z^{(c)} H \Phi_q H). \tag{4}$$

Here the matrices $\Phi_z^{(c)}, H, \Phi_q \in \mathbb{R}^{N \times N}$ are defined as follows. The $(i,j)^{th}$ entry of matrix $\Phi_z^{(c)}$ is computed as $\Phi_{z,i,j}^{(c)} = k(z_i^{(c)}, z_j^{(c)})$, and for matrix $\Phi_q$, the $(i,j)^{th}$ entry is denoted as $\Phi_{q,i,j} = k(q_i, q_j)$. Here $(z_i^{(c)}, q_i)$ and $(z_j^{(c)}, q_j)$ are two samples from $\mathcal{S}_{c,N}$. Matrix $H = I - \frac{1}{N} \mathbf{1} \mathbf{1}^\top$, and notation $tr(\cdot)$ computes matrix trace. In principle, the kernel function $k(\cdot, \cdot)$ can take any legitimate form of characteristic kernel in the RKHS, such as Gaussian kernel $\Phi_{z,i,j}^{(c)} = k(z_i^{(c)}, z_j^{(c)}) = \exp\left(-\frac{\|z_i^{(c)} - z_j^{(c)}\|_2^2}{2\tau}\right)$ or linear kernel. The conditional independence loss sums over all classes is:

$$\mathcal{R} = \frac{1}{C} \sum_c^C \hat{\mathcal{R}}(z, q, c, \theta). \tag{5}$$

In order to facilitate the practical training, we design a specific memory bank architecture in Section 3.3, under which the loss $\mathcal{R}$ in Eq. (5) is penalized in closed form. In the meanwhile, it is crucial to include the conventional softmax classification loss to train for semantically useful features for labeled in-distribution data, where $y_i = c$ is label of $z_i^{(c)}$:

$$\mathcal{L}_{in} = -\frac{1}{CN} \sum_c \sum_i \log \left[ \frac{\exp\left((w^{(c)})^\top z_i^{(c)}\right)}{\sum_{\ell=1}^C \exp\left((w^{(\ell)})^\top z_i^{(\ell)}\right)} \right]. \tag{6}$$

The classifier variable $w^{(c)}$ to be estimated was defined in generative process Eq. 1. The overall loss that backpropagates through the function $f(\cdot; \theta)$ is:

$$\mathcal{L} = \mathcal{L}_{in} + \lambda \mathcal{R}, \tag{7}$$

where $\lambda$ is a pre-defined penalty hyperparameter. The parameters $\theta$ are optimized during the training.

### 3.3 Implementation of condition independence

In contrast to HOOD [39], Conditional-i model particularly incorporates the additional class condition $c$ into loss Eq. (5). Such a new assumption introduces nuanced algorithmic complexity: Computing

$\hat{\mathcal{R}}(\boldsymbol{z}, \boldsymbol{q}, c, \boldsymbol{\theta})$ over all classes $c$ as in Eq. (5) is prohibitively expensive and impractical for batch-wise training. To ensure Eq. (5) stays practically useful, we propose an auxiliary memory bank architecture that allows for convenient end-to-end training. The general idea is to form $C$ number memory banks for computing each class specific matrix $\Phi_z^{(c)}$. During each training iteration, we dynamically maintain and update memory banks $\boldsymbol{V} = \{\boldsymbol{v}_1, \boldsymbol{v}_2, ..., \boldsymbol{v}_c, ..., \boldsymbol{v}_C\}$. All memory banks $\boldsymbol{v}_c$ are of the size $N$ (i.e., $N$ features in each $\boldsymbol{v}_c$) and are formed under a queue structure with "first in first out" policy. Features $\boldsymbol{z}_i^{(c)}$ from the current arriving in-distribution mini-batch is enqueued to each corresponding memory bank $\boldsymbol{v}_c$ indexed by class $c = y_i^{(in)}$, and the oldest sample feature $\boldsymbol{z}_j^{(c)}$ in the queue $\boldsymbol{v}_c$ with $c = y_j^{(in)}$ is removed. During each training iteration, we randomly sample $\hat{C} < C$ classes from $\boldsymbol{V}$, and use the feature values stored in $\boldsymbol{v}_c$ to compute $\Phi_z^{(c)}$ in Eq. (4). We then can compute Eq. (4) with the current arriving batch of OOD feature $\boldsymbol{q}$ of size $N$, at the current $\boldsymbol{\theta}$.

### 3.4 A novel statistical test

During test time inference, a natural choice would be to reuse the independence metric of Eq. (5) for test data. But we unfortunately cannot sample multiple i.i.d. samples to construct $\Phi_q$ during test, since we have no access to test data distribution and $\Phi_q$ would be a constant. We resolve this problem by resorting to a surrogate function that can correctly reflect independence alternatively. As discussed in [22, 62], HSIC independence metric can be further reduced for standardized features if linear kernel is used to construct $\Phi_q$ and $\Phi_z^{(c)}$, yielding the independence metric:

$$tr(\Phi_z^{(c)} \boldsymbol{H} \Phi_q \boldsymbol{H}) = \|\boldsymbol{Z}^\top \boldsymbol{Q}\|_F, \tag{8}$$

where $\boldsymbol{Z} \in \mathbb{R}^{N \times d}$ has features $\boldsymbol{z}_i^\top$ as rows, and $\boldsymbol{Q} \in \mathbb{R}^{N \times d}$ has features $\boldsymbol{q}_i^\top$ as rows. We then follow this thread and modify the Eq. (8) into a novel practical test metric that suits Conditional-i training (different from [39]): during the test, we replace $\boldsymbol{Q}$ in Eq. (8) by a test feature constructed matrix $\boldsymbol{S}_i$, where each row in $\boldsymbol{S}_i \in \mathbb{R}^{N \times d}$ repeats the identical test feature $\boldsymbol{s}_i^\top \in \mathbb{R}^{1 \times d}$. We then concatenate all the training inliers in each $c$ class into each matrix $\boldsymbol{Z}^{(c)} \in \mathbb{R}^{d \times N}$, and compute:

$$t_i = \max_c \frac{1}{N} \|\boldsymbol{Z}^{(c)\top} \boldsymbol{S}_i\|_F. \tag{9}$$

Here $\| \cdot \|_F$ is Frobenius norm. Eq. (9) searches for the inlier class $c$ that returns the maximum conditional dependence with the test feature $\boldsymbol{s}_i$. We determine test feature $\boldsymbol{s}_i$ to be OOD if $t_i < \alpha$ for a predetermined hyperparameter $\alpha$ owing to significant independence with any possible class $c$; and classify it to be inliers if $t_i \geq \alpha$, showing evident dependence with certain inlier class. By ranging across various threshold values $\alpha$ to implement the detection, we obtain the conventional FPR95, AUROC, and AUPR evaluation metrics by definition [28].

## 4 Analysis

The analytic results presented here fully justify Conditional-i's validity as an advantageous OOD detection algorithm. Let $F$ denote the function indexed by parameters $\boldsymbol{\theta}$ such that for a given input $(\boldsymbol{x}_i^{(1)}, \ldots, \boldsymbol{x}_i^{(C)}, \boldsymbol{u}_i)$, $F(\boldsymbol{x}_i^{(1)}, \ldots, \boldsymbol{x}_i^{(C)}, \boldsymbol{u}_i; \boldsymbol{\theta})$ returns output features $(\boldsymbol{z}_i^{(1)}, \ldots, \boldsymbol{z}_i^{(C)}, \boldsymbol{q}_i)$, where $\boldsymbol{z}_i^{(c)} = f(\boldsymbol{x}_i^{(c)}; \boldsymbol{\theta})$ for $c = 1, \ldots, C$, indicating the feature extracted from an incidence in class $c$, and $\boldsymbol{q}_i = f(\boldsymbol{u}_i; \boldsymbol{\theta})$ is the feature extracted from an outlier, and $f$ is defined in Section 3. Our objective Eq. (7) over the parameter $\boldsymbol{\theta}$ then can be viewed as the Lagrangian of the following constrained optimization problem over $F$:

$$\min_F \quad -\frac{1}{CN} \sum_{c=1}^{C} \sum_{i=1}^{N} \log \left[ \frac{\exp((\boldsymbol{w}^{(c)})^\top \cdot \boldsymbol{z}_i^{(c)})}{\sum_{\ell=1}^{C} \exp((\boldsymbol{w}^{(\ell)})^\top \cdot \boldsymbol{z}_i^{(\ell)})} \right] \tag{10}$$

$$\text{s.t.} \quad \{\boldsymbol{z}_i^{(c)} : i = 1, \ldots, N\} \perp\!\!\!\perp \{\boldsymbol{q}_i : i = 1, \ldots, N\}, \forall c \tag{11}$$

$$\|\boldsymbol{z}_i^{(c)}\|_2 = \|\boldsymbol{q}_i\|_2 = 1, \forall i, c \tag{12}$$

where symbol $\perp\!\!\!\perp$ denotes independence between variables.

### 4.1 Condition-i versus HOOD

The distinction between HOOD and Conditional-i approach is made apparent by comparing the HSIC metrics for dependence in the optimization problem. In HOOD, the metric for dependence is $\text{HSIC}_{\text{HOOD}} = tr(\Phi_z \boldsymbol{H} \Phi_q \boldsymbol{H})/(N-1)^2$, where $\Phi_{z,i,j} = k(\boldsymbol{z}_i, \boldsymbol{z}_j)$ with $\boldsymbol{z}_i, \boldsymbol{z}_j$ being samples

from the mixture distribution $p_M(\boldsymbol{z}) = \sum_{c=1}^{C} p_\theta(\boldsymbol{z}^{(\xi)} \mid \xi = c) p(\xi = c)$. On the other hand, the metric for dependence in Conditional-i is $\text{HSIC}_{\text{cond-i}} = \frac{1}{C} \sum_{c=1}^{C} tr(\Phi_z^{(c)} \boldsymbol{H} \Phi_q \boldsymbol{H})/(N-1)^2$, where $\Phi_{z,i,j}^{(c)} = k(\boldsymbol{z}_i^{(c)}, \boldsymbol{z}_j^{(c)})$ are samples from the conditional distribution of inliers given class $\xi = c$. The outlier features that minimize $\text{HSIC}_{\text{HOOD}}$ are those independent of inlier features from the mixture distribution. The outlier features that minimize $\text{HSIC}_{\text{cond-i}}$ are those independent of inlier features from distribution of each of the $C$ classes, and thus also minimize $\text{HSIC}_{\text{HOOD}}$, but the opposite is not necessarily true. A heuristic is given below in the scenario of $d = 1$ and linear kernels: As shown in [62], given the features are standardized, the HSIC metrics can be written as $\|\boldsymbol{z}^\top \boldsymbol{q}\|_F^2$, and thus the minimizer $\boldsymbol{q}$ of Conditional-i is orthogonal (inner product defined by the correlation) to the space spanned by features of all classes $\{\boldsymbol{z}^{(c)} : c = 1, \ldots, C\}$. Minimizers $\boldsymbol{q}$ of HOOD are those orthogonal to the linear combination $\sum_{c=1}^{C} p(\xi = c)\boldsymbol{z}^{(c)}$, which are not necessarily independent of each $\boldsymbol{z}^{(c)}$. In fact, by choosing proper weights, $\boldsymbol{q}$ can be a weighted combination of $\{\boldsymbol{z}^{(c)}\}$ and still be a minimizer of HOOD (see more evidence and illustration in supplementary file).

## 4.2 Inlier and outlier independence guarantee

In terms of the optimum of optimization problem Eq. (10)-(12), we adapt the technique from [43] to support our proof (see supplementary file). But unlike previous efforts as in [43], our approach is anchored in a principled loss function Eq. (10)-(12) with drastically distinct objective, modeling and bounds tailored for OOD detection task.

We firstly introduce the used assumptions and definitions to support our analysis:

**Assumption 1.**
**i)** *Admissible generative functions $\{g^{(c)} : c = 1, \ldots, C\}$ and feature extractors $\{F(\cdot; \boldsymbol{\theta})\}$ are third-order differentiable.*
**ii)** *The Rademacher complexity of the function class $\{F_l^{(c)}(\cdot; \boldsymbol{\theta})\}$ for each $c$ and $l$ is bounded by $R_N$ given $N$ samples.*
**iii)** *Any third order derivative of function $h_l^{(c)} = F_l^{(c)} \circ (g^{(1)}, \ldots, g^{(C)}, g^{(o)})$ is in $[-C_d, C_d]$, where $C_d$ is a finite constant. In addition, $h^{(c)}$ is invertible with respect to $\boldsymbol{w}^{(c)}$.*
**iv)** *There exists a positive constant $\nu$ such that $\sup \|h^{(c)}(\boldsymbol{w}_i^{(1)}, \ldots, \boldsymbol{w}_i^{(C)}, \boldsymbol{\beta}_i) - \boldsymbol{w}^{(c)}\|_2^2 \leq \nu$ almost surely for all $c$.*
**v)** $\sup\{|F_l^c(\boldsymbol{x}_i^{(1)}, \ldots, \boldsymbol{x}_i^{(C)}, \boldsymbol{u}_i)|, |w_l^{(c)}|\} \leq C_f$ *almost surely for all $c$ and $l$, where $C_f$ is a constant.*

Under the above assumptions and definitions, we propose the following theorem:

**Theorem 1.** *Suppose $F$ is a solution of the optimization Eq. (10)-(12). In addition, let $F^{(c)}$ denote the function such that $F^{(c)}(\boldsymbol{x}^{(1)}, \ldots, \boldsymbol{x}^{(c)}, \ldots, \boldsymbol{x}^{(C)}, \boldsymbol{u}) = \boldsymbol{z}^{(c)}$, and let $F_l^{(c)}$ denote the $l^{th}$ element of the output of $F^{(c)}$ for $l = 1, \ldots, d$. Suppose Assumption 1 holds, then the following holds with probability of at least $1 - \delta$: $\forall d_1, d_2 \in \{1, \ldots, d\}$, we have*

$$\mathbb{E}\left[\left\|\frac{\partial F_{d_1}^{(c)}(\boldsymbol{x}^{(1)}, \ldots, \boldsymbol{x}^{(C)}, \boldsymbol{u})}{\partial(\boldsymbol{\beta})_{d_2}}\right\|\right] \leq K C_d^{1/3} \cdot \epsilon^{1/3}, \tag{13}$$

*where $\epsilon = \nu + K_1 R_N + 4C_f^2 \sqrt{\log(1/\delta)/(2N)}$, and $K$ and $K_1$ are positive constants.*

Theorem 1 explains why the independence assumption is beneficial for OOD detection. If we could achieve perfect independence between $\boldsymbol{z}$ and $\boldsymbol{q}$ through the network $f$ conditional on each $\xi = c$, then under mild conditions, we obtain accurate disentanglement between the in-distribution feature $\boldsymbol{z}$ and the *true generative* variable $\boldsymbol{\beta}$ that have *genuinely generated* OOD data, up to a limited error. In other words, the variation in OOD generative $\boldsymbol{\beta}$ does not cause a change in $\boldsymbol{z}$, as literally described in Theorem 1. As the Rademacher complexity increases, the required number of training samples $N$ to achieve accurate independence also increases, in order to effectively bound our disentanglement error between the in-distribution feature and the true generative process of OOD samples.

In this way, Theorem 1 allows us to quantitatively describe how well the inherent generative process of in-distribution data and that of OOD data are disentangled up to an error. If we were using a conventional softmax classifier to train for the in-distribution data, in the presence of the OOD data, the best disentanglement we can achieve is bounded by Theorem 1. This justifies the legitimacy and relevance of our proposed method for the OOD detection task, where the detection goal is to

differentiate the samples belonging to different generative classes from the known in-distributional data. This hopefully may better generalize the training to unseen test OODs as long as the generative process of the test OOD is genuinely different. Theorem 1 only considers the scenario where perfect independence is achieved through penalizing Eq. (4). Certainly, we could further introduce an error term that bounds how much Eq. (4) is deviated from zero. We nevertheless choose to present the most simplified version here for clarity. We defer the more complicated derivation that Eq. (4) is nonzero to future work.

## 5 Experiments

In this section, we evaluate the performance of the Conditional-i method against the state-of-the-art OOD detection methods on both computer vision and NLP (natural language processing) applications. Across all the experiments, the Gaussian kernel is used to compute loss Eq. (5). We evaluate the models under two setups: 1. The training inliers data and training outliers are assumed available, but the labels of training outliers are unknown. 2. The training outliers are no longer available, and we manually generate fake outliers by applying strong augmentation [14] and CutMix [72] on the training inliers. All models are tested on a mixture of test inliers and outliers with the goal of correctly identifying outliers. We compare all methods in terms of FPR95, AUROC, and AUPR metrics [28].

### 5.1 Datasets and baselines

**Datasets. In-distribution training data:** We respectively use the CIFAR-100 and ImageNet1K as the in-distribution training data for image OOD detection tasks. We employ the 20 Newsgroups data as the in-distribution training data for the NLP OOD detection task. **OOD training data:** By following the protocol in [28], we use the 80 Million Tiny Images[2] [61] as the OOD training data when the In-distribution data is CIFAR-100 [32]. We let the ImageNet21K [54] be the OOD training data when ImageNet1K [54] is the in-distribution training data. We define WikiText-2 [46] to be training OOD, when 20 Newsgroups is in-distribution data. **In-distribution test data:** We use the validation data from the corresponding in-distribution dataset as the in-distribution test data as defined in [28]. **OOD test data:** We employ the benchmarks: DTD [13], SVHN [51], Places69 [80], LSUN [71], and CIFAR-10 [32] as the OOD detection test data for computer vision tasks; We use SNLI [7], IMDB [45], Multi30K [17], WMT16 [6], Yelp [2], and EWT [4] as OOD test data for NLP tasks. The specific training in- and out- distribution pairing strategy is (illustrated as inlier training data: OOD test data) CIFAR-100: DTD, SVHN, Places365, LSUN, and CIFAR-10; ImageNet1K: DTD, SVHN, Places365, and LSUN; 20 Newsgroups: SNLI, IMDB, Multi30K, WMT16, Yelp, and EWT.

**Baselines:** The alternative SOTA baselines are K-Base [27], ODIN [38], Mahalanobis distance [34], OE [28], Energy [41], ReAct [59], GradeNorm [30] and HOOD [39]. Here, K-Base [27] corresponds to the model trained through simple softmax loss using in-distribution data, while the OOD detection test is implemented by thresholding the maximum softmax prediction score.

### 5.2 Training architecture and optimization

We use the WideResNet-28-10 [73] and ResNet18 [25] as the training backbone for vision tasks, with the final deep feature dimension $d = 640$ for WideResNet-28-10 and $d = 512$ for ResNet18. We employ the SGD optimizer with $lr = 0.1$, Nesterov Momentum 0.9, for both Table 1 and 2. We use two-layer GRU [10] for producing Table 3 with Adam [31] optimizer. For Table 1 and 2, we adopt $batchsize = 256$ and 512 respectively for in-distribution data. For Table 3, we use $batchsize = 64$ for in-distribution data. Every benchmark method in Table 1 and 2 is trained for 100 epochs w.r.t. the size of the in-distribution training data. For NLP experiments, we train 2-layer GRUs [11] for 5 epochs for Table 3 (See supplementary material for details). We train each algorithm on 1 P40 GPU. In contrast to the sampling as in [39] and [28] which sequentially sample random blocks of data from 80 Million Tiny Images, we firstly randomly shuffle the training dataset before sampling. The shuffling is expected to eliminate the sampling bias owing to the inherent correlation between the sample ordering and the categories. We found that such shuffling has improved all the baselines. We believe we have tuned all the baseline models to our best. By following the protocol in [28], we

---

[2]This dataset seems to be suspended recently. We therefore will supplement the result of Table 1 by training Conditional-i on ImageNet22K/300K Random Images [28] (A subset of 80 Million Tiny Images) instead, and shortly release the results at code link https://github.com/OODHSIC/conditional-i.

Table 1: Averaged OOD detection performance. Performances on each specific test dataset are in supplementary material. $\mathcal{D}_{in}^{(tr)}$ = CIFAR-100, $\mathcal{D}_{ood}^{tr}$ = 80 Million Tiny Images, $\mathcal{D}_{ood}^{tst}$ = {DTD, SVHN, Place365, LSUN, CIFAR-10}. "True Out. Expo." means known outlier training data exposure.

| $\mathcal{D}_{tr,in}$ | Method | True Out. Expo. | FPR95(%)↓ | AUROC(%)↑ | AUPR(%)↑ |
|---|---|---|---|---|---|
| **CIFAR-100** (WideResNet) | K-Base [27] | no | 65.84±0.7 | 74.76±0.3 | 33.25±0.4 |
| | ODIN [38] | no | 62.64±1.5 | 79.06±0.8 | 38.46±1.2 |
| | Mahalanobis [34] | no | 53.77±1.4 | 82.72±0.3 | 48.39±0.7 |
| | Energy [41] | no | 62.25±1.2 | 79.04±0.4 | 35.79±0.6 |
| | ReAct (+Energy) [59] | no | 59.37±0.6 | 79.90±0.2 | 36.62±0.5 |
| | OE-generative [28] | no | 53.29±0.6 | 83.44±0.3 | 51.25±0.4 |
| | HOOD-generative [39] | no | 54.91±0.8 | 83.12±0.4 | 55.18±1.0 |
| | **Conditional-i-generative (ours)** | no | **51.09±1.4** | **83.74±0.5** | **56.69±1.6** |
| | Energy [41] | yes | 33.07±0.2 | 89.57±0.1 | 56.92±0.4 |
| | OE [28] | yes | 33.41±1.0 | 89.75±0.4 | 57.74±1.2 |
| | HOOD [39] | yes | 33.20±0.6 | 89.81±0.2 | 59.18±1.0 |
| | HOOD-bank | yes | 32.97±0.3 | 89.87±0.2 | 60.16±0.6 |
| | **Conditional-i (ours)** | yes | **32.66±0.2** | **90.03±0.1** | **61.48±0.5** |

Table 2: Averaged OOD detection performance. Performances on each specific test dataset are provided in supplementary material. $\mathcal{D}_{in}^{tr}$ = ImageNet-1K, $\mathcal{D}_{ood}^{tr}$ = ImageNet-21K, $\mathcal{D}_{ood}^{tst}$ = {DTD, SVHN, Place365, LSUN}, "True Out. Expo." means known outlier training data exposure.

| $\mathcal{D}_{tr,in}$ | Method | True Out. Expo. | FPR95(%)↓ | AUROC(%)↑ | AUPR(%)↑ |
|---|---|---|---|---|---|
| **ImageNet1K** (ResNet18) | K-Base [27] | no | 66.05±1.2 | 72.98±0.5 | 32.93±0.9 |
| | ODIN [38] | no | 52.18±0.8 | 80.41±0.4 | 44.53±0.5 |
| | Mahalanobis [34] | no | 75.80±2.6 | 61.10±0.9 | 23.79±1.6 |
| | Energy [41] | no | 54.15±0.8 | 78.73±0.3 | 42.36±0.7 |
| | ReAct (+Energy) [59] | no | 54.59±0.8 | 78.96±0.3 | 42.94±0.8 |
| | OE-generative [28] | no | 63.43±1.8 | 74.11±0.5 | 38.13±2.0 |
| | HOOD-generative [39] | no | 52.28±0.9 | 80.16±0.5 | 44.38±0.7 |
| | **Conditional-i-generative (ours)** | no | **50.20±0.5** | **81.20±0.2** | **44.55±0.6** |
| | Energy [41] | yes | 48.90±0.5 | 84.56±0.3 | 46.66±1.0 |
| | OE [28] | yes | 57.32±1.6 | 81.47±0.6 | 45.37±0.8 |
| | HOOD [39] | yes | 46.30±0.7 | 84.28±0.2 | 50.50±1.2 |
| | HOOD-bank | yes | 48.42±0.6 | 84.67±0.4 | 47.31±0.8 |
| | **Conditional-i (ours)** | yes | **42.84±0.9** | **86.79±0.2** | **55.00±0.6** |

report the average performance by using the consistent hyperparameters, without tuning for each specific test dataset. The detailed results of each specific test can be found in the supplementary file.

## 5.3 Benchmark results

**Image OOD detection tasks.** We first present the OOD detection results in Table 1, where the CIFAR-100 dataset serves as in-distributional data, with 80 Million Tiny Images being the OOD training dataset. DTD, SVHN, Places365, LSUN, and CIFAR-10 are tested as OOD data. We tuned all the compared methods based on the validation set defined in [28]. It manifests from Table 1 that Conditional-i generally outperforms all the existing SOTA OOD methods in terms of FPR95, AUROC, and AUPR metrics. Particularly, Conditional-i surpasses our earlier HOOD method by an evident margin, justifying the unique advantage of the Conditional-i model as discussed in 4.1. We found that the Conditional-i approach is especially efficient in differentiating near OOD samples, such as distinguishing CIFAR-10 from CIFAR-100, while it also provides decent performance against large domain gaps, e.g., CIFAR-100 vs. DTD. In Table 2, where ImageNet1K is inlier and ImageNet-21K is outlier, the performance of Conditional-i is even further improved when finer classes of OOD training data are used for training. This verifies that, by capitalizing on the conditional-independence assumption, Conditional-i exploits unique constraints when searching for the OOD feature parameterization, and henceforth shows advantageous generalization ability under the finer classes' OOD discrimination. This also best supports the relevance of our Theorem 1, that Conditional-i can distinguish inliers and outliers depending on the true generative process of test samples given their class conditions.

**Training without access to known true OOD training data.** Proceeding further, we compare all models when we lose access to the oracle training OOD data. In this scenario, we use generated fake OOD training data by applying strong augmentation [14] and CutMix [72] (for CV tasks) or EDA [68] techniques (for NLP tasks) on each in-distributional data (see supplementary material). Each "XX" baseline trained with fake OOD training data is referred to as "XX-generative" (e.g., Conditional-i-generative). As displayed in Table 1 and 2, Conditional-i-generative performs relatively worse than the vanilla Conditional-i model, owing to the absence of true OOD data. However, Conditional-i-generative not only beats all SOTA models without OOD training data exposure, including K-Base, ODIN, Mahalanobis distance, Energy, and ReAct(+Energy), but also exhibits evident gain over other exposure methods, i.e., OE-generative and HOOD-generative. Here, ReAct(+Energy) corresponds to the ReAct [59] method based on Energy [41].

Table 3: Averaged OOD detection performance on NLP tasks. $\mathcal{D}_{in}^{tr}$ = 20 Newsgroups, $\mathcal{D}_{ood}^{tr}$ = WikiText-2, $\mathcal{D}_{ood}^{tst}$ = {SNLI, IMDB, Multi30K, WMT16, Yelp, EWT}.

| Method | True Out. Expo. | FPR95(%)↓ | AUROC(%)↑ | AUPR(%)↑ |
|---|---|---|---|---|
| K-Base [27] | no | 52.84±3.1 | 84.27±1.7 | 41.41±6.2 |
| ODIN [38] | no | 46.71±3.5 | 86.90±1.2 | 50.98±5.1 |
| Mahalanobis [34] | no | 70.09 ±5.5 | 65.37±2.5 | 15.70±4.3 |
| Energy [41] | no | 41.00±4.1 | 89.95±1.6 | 54.64±7.4 |
| ReAct (+Energy) [59] | no | 40.24±4.3 | 90.21±1.6 | 55.28±7.8 |
| GradeNorm [30] | no | 39.26±1.4 | 85.36±0.2 | **79.02±0.6** |
| OE-generative [28] | no | 49.89±1.1 | 86.20±0.7 | 61.95±1.4 |
| HOOD-generative [39] | no | 37.92±1.4 | 92.51±0.4 | 78.66±1.2 |
| **Conditional-i-generative (ours)** | no | **33.17±1.1** | **93.12±0.6** | 78.94±1.0 |
| Energy [41] | yes | 2.79±0.8 | 99.25±0.2 | 93.67±2.0 |
| OE [28] | yes | 2.99±0.8 | 99.16±0.1 | 94.74±0.3 |
| HOOD [39] | yes | 2.03±0.4 | 99.45±0.1 | 95.63±1.0 |
| HOOD-bank | yes | 1.83±0.3 | 99.44±0.1 | 95.87±1.2 |
| **Conditional-i (ours)** | yes | **1.52±0.3** | **99.56±0.1** | **96.45±1.6** |

Table 4: Ablation on the number of classes $C$ for computing Eq. (5).

| $\hat{C}$ | 1 | 5 | 10 | 100 | 1000 |
|---|---|---|---|---|---|
| FPR95 | 47.08 | 47.18 | 46.83 | 45.43 | **42.84** |
| AUROC | 84.98 | 85.14 | 85.63 | 85.60 | **86.79** |
| AUPR | 47.42 | 47.73 | 49.87 | 49.24 | **55.00** |

Table 5: Ablation against in-distribution batchsize $N$.

| N | 16 | 32 | 64 | 128 | 256 |
|---|---|---|---|---|---|
| FPR95 | 33.37 | 34.02 | 33.94 | 33.02 | **32.66** |
| AUROC | 87.43 | 89.79 | 89.91 | 89.95 | **90.03** |
| AUPR | 47.30 | 57.28 | 61.09 | 61.30 | **61.48** |

**NLP OOD detection tasks.** We observe from Table 3 that Conditional-i offers the strong OOD detection performance on texts across all the evaluation metrics. The advantage of Conditional-i is particularly evident in the AUPR metric, that the Conditional-i model is more than $1\%$ better than the HOOD model, and is $2\%$ better than the OE method. Note although GradeNorm [30] shows superior performance in AUPR metric when training OOD is unavailable, it nevertheless performs poorly in terms of the other metrics, showing the unique benefit of Conditional-i on NLP applications.

## 5.4 Ablation study

**Ablation on memory bank architecture**. The proposed memory bank architecture is necessary and is exclusively designed to facilitate the Conditional-i training. However, to explain the advantage of conditional independence modeling hypothesis against HOOD, it is crucial to ablate the effect of the additional memory bank that is absent in HOOD training. We refer to this new baseline as HOOD-bank, which is constructed with the identical memory bank architecture of Conditional-i. The critical difference here is that each 1 out of $C$ memory bank for HOOD-bank is instead filled with random in-distribution data regardless of their class category. Each arriving in-distribution data randomly chooses 1 out of $C$ memory banks with a probability $1/C$ and joins the corresponding queue. We compute the batch-wise loss function as in Eq. (5) by ranging across $\hat{C}$ number of memory banks of inliers data, where $\hat{C} < C$ classes are sampled from $C$ total memory banks as defined in Section 3.3. In this way, samples in $\Phi_z^{(c)}$ are no longer sampled i.i.d. by conditioning on the class variable $c$ for HOOD-bank. Rather, samples in each bank follow the marginal distribution of $p(\boldsymbol{z}|\boldsymbol{\theta})$ by margining out the classes. Tables 1, 2, 3 clearly demonstrate that, the memory bank architecture alone, i.e., HOOD-bank method, can hardly boost HOOD. The advantage of Conditional-i is mainly attributed to the new conditional latent model. This result directly justifies Section 4.1.

**Ablation on number of classes** $\hat{C}$. We investigate the impact of sampled class number $\hat{C}$ included for computing Eq. (5), by following the procedure in Section 3.3. The training setup is the same as for producing Table 2. We still keep using the total number of classes present in the in-distributional data to train the inlier classifier through Eq. (6). In the meanwhile, we reduce the number of classes included in Eq. (5) to $\hat{C} < C$ and examine the regularizing impact of the additional condition on the OOD detection performance. As shown in Table 4, as more in-distributional data join the loss Eq. (5), the network gradually learns to shrink the subspace in which the OOD training sample can live in, so that the OOD features respect the independence condition posed by each in-distributional class. While we increase the value of $\hat{C}$, the performance keeps improving across all the metrics, until all classes are included.

**Ablation on in-distributional batchsize** $N$. We vary the in-distributional data batch size during training, with the training setup identical to Table 1. Since computing Conditional-i loss involves the estimator construction as in Eq.(4), the number of samples used to compute the kernel matrix $\Phi_z^c$ and $\Phi_q$ would affect the estimator bias in terms of the actual independence. As displayed in Table 5, the OOD detection performance significantly drops when the in-distribution batch size is below 32, owing to the increased estimator bias introduced through Eq. (4).

**Ablation on weight** $\lambda$. The weight $\lambda$ balances the classification loss and the strength of the conditional independence loss. When $\lambda = 0$, Conditional-i degenerates to the K-Base method. When $\lambda = 0.03$, the OOD detection performance reaches a sweet point owing to the inclusion of the kernel based

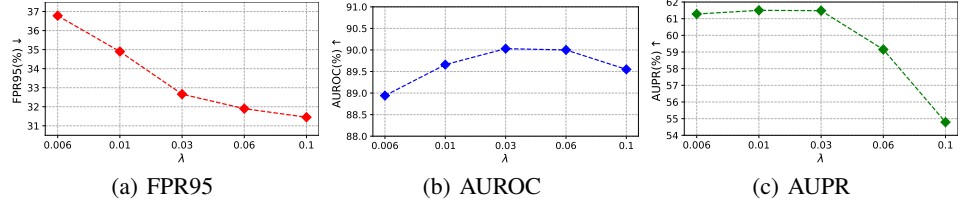

(a) FPR95        (b) AUROC        (c) AUPR

Figure 1: Ablations against weight $\lambda$.

Table 6: Ablation study of applying different test method on different training models.

| Train (Test) | FPR95(%)↓ | AUROC(%)↑ | AUPR(%)↑ |
|---|---|---|---|
| K-Base (SFM) | 66.05 | 72.98 | 32.93 |
| K-Base (Cond-i) | 70.69 | 69.31 | 37.48 |
| Conditional-i (SFM) | 56.76 | 81.20 | 48.37 |
| Conditional-i (Condi-i) | **42.84** | **86.79** | **55.00** |

Table 7: Ablation study against augmentation for Conditional-i-generative.

| Train (Test) | FPR95(%)↓ | AUROC(%)↑ | AUPR(%)↑ |
|---|---|---|---|
| K-Base | 66.05 | 72.98 | 32.93 |
| +Cutmix | 61.45 | 75.77 | 42.17 |
| +RandAug | 56.15 | 79.31 | 42.47 |
| +RandAug+Cutmix | **50.20** | **81.20** | **44.55** |

independence calculation. However, beyond $\lambda = 0.03$, the performance starts to drop again, partially because this overemphasized independence starts to hurt the in-distribution semantic feature extraction that relies on the accuracy of inlier classification. Learning independence from the noisy semantics having high classification loss would then lose the benefit.

**Ablation on test method**. Note our proposed test method Eq. (9) was originally motivated from the training objective Eq. (7) which exclusively couples with Conditional-i training loss. Nevertheless, it still worths trying testing the Conditional-i test/train alone independently. In the Table 6, we use Train (Test) to indicate the model trained with "Train" method and tested with "Test" method. Other training details are the same as Table 2. We apply the Conditional-i test on the model trained via conventional cross entropy loss, which is denoted as K-base (Cond-i). In contrast to the model K-base (SFM) which employs a softmax score (SFM) for test, K-base trained model may hardly benefit from the new test. In the meanwhile, we also observe that Conditional-i (SFM) may hardly beat Conditional-i (Cond-i), showing the critical role of coupled train/test pair. This again verifies that Conditional-i is a holistic procedure with principled motivation and theoretical support.

**Ablation on augmentation for Conditional-i-generative**. This section evaluates the impact of applying different augmentation for the proposed Conditional-i-generative model. The ablation study is shown as below (other training details are the same as Table 2). In Table 7, "+Cutmix" indicates the Cutmix augmentation [72] and "+RandAug" indicates the strong augmentation [14]. As can be seen from Table 7, strong augmentation along with the Cutmix (i.e., +RandAug+Cutmix) would lead to the best results, whereas other augmentation methods would also help Conditional-i to surpass SOTA methods.

## 6 Conclusion and Broader Impact

We propose a novel OOD detection approach referred to as Conditional-i. Inspired by the ICA techniques, Conditional-i is shown to be effectively penalizing the deep feature dependency between inliers and outliers. Conditional-i models conditional independence by incorporating the condition information $c$ into the probabilistic model. Such independence can be explicitly penalized through the use of the HSIC metric. We constructed an efficient memory bank architecture critical for the practical implementation of the method in an end-to-end training manner. Empirical evidence justifies the superior performance of the proposed method on both computer vision and natural language processing benchmarks. We believe OOD detection is an important task regarding AI safety. The accuracy of OOD detection directly impacts the reliability of many AI applications, including healthcare, anomaly detection, autonomous driving, and others. The negative impact is that the development of the Conditional-i method is subject to the limitations of current SOTA scores we can achieve, which may limit the actual safety of the AI applications. Whereas Conditional-i achieves the SOTA performance, another limitation though, is that Conditional-i needs to be trained on (fake) OOD training data, that might potentially encounter generalization issues, by virtue of [3].

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
