# OpenReview forum: "Out-of-Distribution Detection via Conditional Kernel Independence Model"
_NeurIPS.cc/2022/Conference — NeurIPS 2022 Accept_

### Official Review · Reviewer_LynZ · 2022-06-15

**Rating:** 5
**Confidence:** 5
**Soundness:** 2 fair
**Presentation:** 1 poor
**Contribution:** 2 fair

**Summary:**

This paper proposes a novel method to address OOD detection by independent component analysis (ICA) techniques. This novel method named Conditional-i builds upon the probabilistic formulation, and applies the Hilbert-Schmidt Independence Criteria that offers a convenient solution for optimizing variable dependencies.

Basic Idea:
For different ID class-conditional distributions, the paper uses MMD to estimate the dependence between ID for c-class and OOD. Main formulas are in Eqs. 3 and 4.

Theorem 1 explains why the independence assumption is beneficial for OOD detection.

Experiments have shown Conditional-i is better than some classical methods.


**Questions:**

(Major) 1. Poor writing. Many notations are not standard. I spend many times to understand your notations, like lines 127-128. What is p_{theta}? it is a mixture of ID and OOD distributions? You should revise your notations to be more readable.

(Major) 2. In Eq 11, do you assume that the number of OOD data is N, which is same with the number of c-class ID data?

(Major) 3. If we have not any OOD data, how you train your model? It is really difficult to find where you have described it. Could you tell me which line you have introduced it?

(Major) 4. Please compare the SOTA method Gradenorm (On the importance of gradients for detecting distributional
shifts in the wild.)

(Minor) 5. The idea of  HOOD is similar with this paper. You use an extremely similar idea to submit two papers?



**Limitations:**

The authors adequately addressed the limitations and potential negative societal impact of their work

**Strengths And Weaknesses:**

Strengths:
1. Experiments show Conditional-i is better than some classical methods.
2. The idea that uses independent component analysis (ICA) techniques seems to be novel.


Weakness:
(Major) 1. Poor writing. Many notations are not standard. I spend many times to understand your notations, like lines 127-128. What is p_{theta}? it is a mixture of ID and OOD distributions? You should revise your notations to be more readable.

(Major) 2. In Eq 11, do you assume that the number of OOD data is N, which is same with the number of c-class ID data?

(Major) 3. If we have not any OOD data, how you train your model? It is really difficult to find where you have described it. Could you tell me which line you have introduced it?

(Major) 4. Please compare the SOTA method Gradenorm (On the importance of gradients for detecting distributional
shifts in the wild.)

(Minor) 5. The idea of  HOOD is similar with this paper. You use an extremely similar idea to submit two papers?

---

> ### Author Response · Authors · 2022-08-02
> **Response to Reviewer LynZ (Part 3/3)**
>
> (continued from Part 2/3)
>
> **Q5**: The idea of HOOD is similar with this paper. You use an extremely similar idea to submit two papers?
>
> **Ans-Q5**: We would like to emphasize that paper [2] was an earlier trial that conveys very preliminary trails and limited results that were submitted earlier to another venue (under review), which in certain cases HOOD [2] would probably fail in discriminating OOD samples. In this regard, please also refer to analysis in Section 4.1 in our Conditional-i paper, and also see supplementary material line 43-55. We were happy to share that earlier idea [2] with the community, whereas we think Conditional-i should not to be criticised for continuously challenging the usefulness of the HSIC hypothesis.
>
>
> In fact, the Conditional-i paper points out many limitation of HOOD, while we associate rigorous proofs to support these claims. We also encourage the reviewer to re-consider the details given in Section 4.1, which also manifests itself that the optima of HOOD and Conditional-i would be drastically different. Our independence constraints Eq. (11) are crucial in disentangling the extracted OOD features from the in-distribution features and generators. Constraining on the independence between the OOD and each in-distribution class's features guarantees the OOD feature extractor is an invertible function of the OOD generator alone without mixing with in-distribution generators.
>
> Here we provide an example to demonstrate the difference in solution between the Conditional-i and HOOD methods. With results of Sections 3.4 and 4.1 in the main text, in the scenario of $d=1$ and linear kernel, $HSIC_{hood} = \vert{\boldsymbol{Z}}^{\top}\boldsymbol{Q} \vert_F$, where $\boldsymbol{Z}$ denotes the vector of samples from the mixture distribution $p_M(\boldsymbol{z}) = \sum_{c=1}^C p_\theta(\boldsymbol{z}^{(\xi)}\mid \xi = c) p(\xi = c)$ and $\boldsymbol{Q}$ denotes the samples of out of distribution features $\boldsymbol{q}$. When the sample size is large, $HSIC_{hood}$ is approximately the Frobenius norm of $N$ times of the correlation between the out of distribution feature and $\boldsymbol{z}$ from the mixture distribution. Minimizers $\boldsymbol{q}$ of $HSIC_{hood}$ are those orthogonal (inner product defined by the correlation) to the linear combination $\sum_{c= 1}^C p(\xi = c) \boldsymbol{z}^{(c)}$, where $\boldsymbol{z}^{(c)}$ follows the conditional distribution of $\boldsymbol{z}$ given class $\xi = c$. Suppose the total number of classes $C=2$, $p(\xi = 1) = p(\xi = 2) = 1/2$, and the correlation between $z^{(1)}$ and $z^{(2)}$ is omissible. Then $\boldsymbol{q} = \boldsymbol{z}^{(1)}/\sqrt{2} - \boldsymbol{z}^{(2)}/\sqrt{2}$ minimizes $HSIC_{hood}$ but is not orthogonal of each $\boldsymbol{z}^{(c)}$. On contrary, the minimizer of $HSIC_{cond-i} $ is required to be orthogonal of each $\boldsymbol{z}_i^{(c)}$. This is a perfect example demonstrating the capability of the proposed Conditional-i method in disentangle out-of-distribution features from the in distribution features and generators, when HOOD fails.
>
> By virtue of the analysis above, Conditional-i shows strong empirical superiority over HOOD, by successfully (and theoretically) circumventing these limitations and failure cases (through construction of novel memory bank structures). Note that this paper also considers using a brand new generative process hypothesis that were especially tailored for Conditional-i, which also supports the construction of Theorem 1.  Such specific generative modeling does not transfer to HOOD [2] owing to its special assumption set, and HOOD also did not enjoy similar guarantee bounds of Conditional-i. Perhaps the last but not least, the test of conditional-i is distinct from that used in HOOD in principle. HOOD [2] introduces a simple linear correlation test that measures between the features that does not sufficiently reflect the independence. However, the new test here was particularly motivated and suitable to couple with the modeling of Conditional-i training.
>
> We sincerely appreciate it if Reviewer LynZ may kindly reconsider the above points during this rebuttal.

---

> > ### Comment · Reviewer_LynZ · 2022-08-03
> > **To Response**
> >
> > I have read your response.
> >
> > I will increase your score t0 5. But it does not mean that I like your response.
> >
> > I still suggest you to revise your notation according to classical machine learning book but not statistical inference.
> >
> > Maybe you are familiar with the notations in statistical inference, but as I know, not so many machine learning researchers are familiar with that, especially in the OOD detection community.  For reviewers, they have not so many time to read new books and re-learn novel notations for your paper. Additionally, you also can add the explaination about important notations in appendix, but you have not done it and even in the response, you also have not such response about restating main notations in appendix.
> >
> > So if you want your paper can be accepted, please refine your writing and pay more attention on it.

---

> > > ### Author Response · Authors · 2022-08-03
> > > **Thanks for your response**
> > >
> > > Dear Reviewer LynZ,
> > >
> > > We sincerely appreciate your valuable time on our response! We are very glad to see some of our answers have clarified the concerns. In the meanwhile, we also apologize for the confusions on the notations, and we promise to improve the definition and the presentation in the revision. We will make sure to include more explanations and clarifications both in the main text and in the appendix.
> > >
> > > Thanks again for your kind suggestions!

---

> ### Author Response · Authors · 2022-08-02
> **Response to Reviewer LynZ (Part 2/3)**
>
> (continued from Part 1/3)
>
> **Q2**: In Eq 11, do you assume that the number of OOD data is N, which is same with the number of c-class ID data?
>
> **Ans-Q2**: Yes, for the sake of batchwise training, we make sure during each iteration, the number of OOD data is the same as the batchwise c-class ID data, so that we can compute Eq. (4) and Eq. (5).
>
> **Q3**: If we have not any OOD data, how you train your model? It is really difficult to find where you have described it. Could you tell me which line you have introduced it?
>
> **Ans-Q3**: Lines 334-342 in main paper (Section started with bolded "Training without access to known true OOD training data") exactly described how we implement the OOD detection when training OOD data is absent. Further details of how we generated the fake OOD samples can also be found in supplementary file line 56-73 (see Section 4 Conditional-i-generative in supplementary file). In a nutshell, we generate fake OOD training samples through applying strong augmentation on original In-distribution data. More experiments and discussions on Conditional-i-generative can also be found in the response to other Reviewers.
>
> **Q4**: Please compare the SOTA method Gradenorm (On the importance of gradients for detecting distributional shifts in the wild.)
>
> **Ans-Q4**: Thanks a lot for providing more useful references. During this short time of rebuttal, we made our best to tune for this mentioned Gradenorm method and we obtain the following score comparisons. Under our training set up and training architecture, Conditional-i strongly outperforms Gradenorm.
>
> The scores of Gradenorm on different training data are:
> | **In-Dataset**               | **FPR95**     | **AUROC**     | **AUPR**      |
> |--------------------------|-----------|-----------|-----------|
> | CIFAR-100                | 88.78     | 59.10     | 53.04     |
> | IN1K                     | 65.08     | 80.06     | 77.14     |
> | 20NG                     | 39.26     | 85.36     | 79.02     |
>
> (To be continued in **Part 3/3**).

---

> ### Author Response · Authors · 2022-08-02
> **Response to Reviewer LynZ (Part 1/3)**
>
> We appreciate the constructive comments from Reviewer LynZ and the time Reviewer LynZ spent on reviewing the paper. We also thank Reviewer LynZ's kind patience in engaging in this rebuttal. Let us address some key confusions and concerns here. We hope the information here may further supplement our main submission and justify the soundness of Conditional-i.  We will include the primary results and discussion here in the main paper.
>
> **Q1**: Poor writing. Many notations are not standard. I spend many times to understand your notations, like lines 127-128. What is $p_{\theta}$? it is a mixture of ID and OOD distributions? You should revise your notations to be more readable.
>
> **Ans-Q1**: We do strictly follow the canonical probability notations and definitions in the machine learning and statistics community. As it was described in paper in line 132, $p_{\theta}$ defines a probability density parameterized by $\theta$. Take for instance, $p_{\theta}({\boldsymbol{z}}^{(\xi)}, {\boldsymbol{q}}|\xi=c)$ denotes the joint probability density of IN and OOD feature variables $({\boldsymbol{z}}^{(\xi)}, {\boldsymbol{q}})$ conditional on the event $\{\xi=c\}$, where the density depends on the value of the parameter $\theta$. Similar notation usage of $p_{\theta}$ were considered standard protocol and can be conveniently found in textbook such as [R4A] [R4B], or in highly influential papers such as VAE [R4C].  We hope these references are helpful in easing the symbols understanding in our paper.
>
>
> Here, $p_{\theta}({\boldsymbol{z}}^{(\xi)}, {\boldsymbol{q}}|\xi=c)$ is not a mixture distribution. We did mention the term "mixture distribution" in a different context though, i.e., line 217-218 to make a distinction between the vanilla HOOD method and the proposed Conditional-i method. The HOOD requires only the independence between the OOD feature and the mixture of IN samples from all classes, while Conditional-i requires the independence of the OOD with each of the IN classes. We then examine in depth the benefit of the Cond-i constraints, see, Sections 4.1. We appreciate the reviewer for raising this important distinction of concepts.
>
> **References**
>
> [R4A] George G. Roussas. An introduction to probability and statistical inference. 2003. Academic Press.
>
> [R4B] Casella, George; Berger, Roger L. Statistical Inference (Second ed.). Thomson Learning. pp. 34–37. 2002.
>
> [R4C] D.P. Kingma, Max Welling. Auto-Encoding Variational Bayes. 2013.
>
> (To be continued in **Part 2/3**).

---

### Official Review · Reviewer_1WG2 · 2022-07-10

**Rating:** 6
**Confidence:** 4
**Soundness:** 3 good
**Presentation:** 3 good
**Contribution:** 3 good

**Summary:**

The paper proposes a novel training method for improving the out-of-distribution (OOD) detection performance of a classification model. It is motivated by the hypothesis that the mutual information (dependency) between OOD data and in-distribution (ID) data should be small. Specifically, the proposed method relies on the concept of class conditional independence between ID features and OOD features and utilities maximum mean discrepancy (MMD) to enforce this condition, leading to a composite loss function and training strategy. The method evolves into two variants where one requires unlabeled OOD data and the other uses generated OOD data through strong data augmentation.

**Questions:**

1. Could the authors comment on the accuracy of this method?
2. Could the authors comment on the sensitivity of the method using different OOD data during training or different augmentation strategies?


**Limitations:**

The authors addressed potential negative impact in the paper.

**Strengths And Weaknesses:**

Strength
1. The paper is very written, especially on addressing the similarity to a contemporary work and descriptions of small implementation details that are important for the success of this method in practice.
2. The method is thetically motivated and empirically verified on large datasets including both CV and NLP tasks.
3. The paper includes ablation study on various aspects of the algorithm such as the number of classes for enforcing independence during training and the hyperparameter for adjusting the contribution of the loss component for enforcing independence.

Weakness:
1. As with all methods requiring OOD data during training, the proposed method necessarily entails overhead on storing training-data, sampling/generating OOD data and additional computation such as the Gaussian Kernel.
2. The performance against a specific type of OOD data of the proposed method might be highly dependent on the type of OOD exposure or augmentation used during training.
3. It’s not clear how the proposed composite loss function would affect the discriminativeness (accuracy) of the model on in-distribution data.

---

> ### Author Response · Authors · 2022-08-02
> **Response to Reviewer Reviewer 1WG2 (Part 2/2)**
>
> (continued from Part 1/2)
>
> **Q1**: Could the authors comment on the accuracy of this method?
> The top 1 accuracy on the in-distribution data is reported as below. As can be seen, we indeed sacrifice a bit/neglecatble classification accuracy on the in-distribution data. However, it seems this is a worthwhile trade-off, since Conditional-i can effectively boost the OOD detection performance to a large extent at the merely marginal cost of classification accuracy.
>
> **Ans-Q1**: We illustrate the corresponding top-1 acc on in-distribution data for Table 1,2,3 main paper as follows:
>
> | **Methods**        | **top-1 (CIFAR-100)** | **top-1 (IN1K)** | **top-1 (20NG)** |
> |----------------|-------------------|--------------|--------------|
> | K-base         | **80.76**         | 54.01        | 26.31        |
> | OE             | 78.33             | **53.55**    | 28.21        |
> | Energy         | 77.72             | 52.14        | 27.91        |
> | HOOD           | 77.83             | 53.10        | 28.30        |
> | Conditional-i  | 77.21             | 52.87        | **28.36**    |
>
> As can be seen from the table above, we indeed occasionally sacrifice a bit classification accuracy on the in-distribution data. However, it seems this is a worthwhile trade-off, since Conditional-i can effectively boost the OOD detection performance to a large extent at marginal cost of classification accuracy.
>
> **Q2**: Could the authors comment on the sensitivity of the method using different OOD data during training or different augmentation strategies?
>
> **Ans-Q2**: We used two different augmentations, with the motivation to verify the flexibility and robustness of Conditional-i on different augmentation methods. However, it is a good suggestion to also ablate out the effectiveness of the augmentation types. During the rebuttal time, we were able to re-train the ImageNet1K experiment (Table 2 in the main paper) using different augmentations. We use "+Cutmix" to indicate the Cutmix augmentation whereas we use "+RandAug" to denote the strong augmentation.  The ablation study is shown as below (Other training details remain the same as Table 2 in main paper):
>
> | **Methods**         | **FPR95**     | **AUROC**     | **AUPR**      |
> |-----------------|-----------|-----------|-----------|
> | K-base          | 66.05     | 72.98     | 32.93     |
> | +Cutmix         | 61.45     | 75.77     | 42.17     |
> | +RandAug        | 56.15     | 79.31     | 42.47     |
> | +RandAug+Cutmix | **50.20** | **81.20** | **44.55** |
>
> As can be seen from the table above, strong augmentation along with the Cutmix (i.e., +RandAug+Cutmix) would lead to the best results, whereas other augmentation methods would also help Conditional-i to beat SOTA methods such as ReACT and Energy.

---

> ### Author Response · Authors · 2022-08-02
> **Response to Reviewer 1WG2 (Part 1/2)**
>
> We sincerely appreciate the kind comments from Reviewer 1WG2. We also thank Reviewer 1WG2 for recognizing the merits in our paper and your kind patience in engaging in this rebuttal. Let us address some key concerns here. We hope the information here may further supplement our main submission and justify the soundness of Conditional-i.  We will include the primary results and discussion here in the main paper.
>
> **W1**: As with all methods requiring OOD data during training, the proposed method necessarily entails overhead on storing training-data, sampling/generating OOD data and additional computation such as the Gaussian Kernel.
>
> **Ans-W1**: During training, we generate the fake OOD training data on the fly, which incurs similar computational cost as conventional augmentation, and would not incur much difficulties or GPU memory issues during training. Honestly, the kernel method indeed would incur some computation complexity in $O(N^{3})$. However, this seems to be somewhat an accuracy-complexity trade-off that worths the trail. Besides, there are also methods that focus on relieving the computational cost of kernel methods [R3A], which can effectively reduce the computational cost of kernel method to linear cost. We are quite interested in further examining the efficiency of such kind of solution [R3A] on Conditional-i, whereas this seems not the focus of this paper though.  But we will make sure to include this part of discussion in the paper.
>
> **Reference:**
>
> [R3A] A. Rahimi and B. Recht. “Random Features for Large-Scale Kernel Machines.” NeurIPS. 2007
>
> **W2**: The performance against a specific type of OOD data of the proposed method might be highly dependent on the type of OOD exposure or augmentation used during training.
>
> **Ans-W2**: Please see our Ans-Q2 below for responding to this point.
>
> **W3**: It’s not clear how the proposed composite loss function would affect the discriminativeness (accuracy) of the model on in-distribution data.
>
> **Ans-W3**: Please see the answer Ans-Q1 below for this answer.
>
> (To be continued in **Part 2/2**).

---

### Official Review · Reviewer_62WK · 2022-07-11

**Rating:** 6
**Confidence:** 4
**Soundness:** 2 fair
**Presentation:** 2 fair
**Contribution:** 2 fair

**Summary:**

This paper proposes a conditional independence model called Conditional-I for OOD detection. The authors introduce an integrated loss function to reduce the dependence between inliers features of each class and outlier features, so as to distinguish the OOD samples by a relevant independence measure during testing. The authors provide theoretical guarantees of the conditional independence hypothesis. The experiments demonstrate the better performance of the proposed method on CV and NLP tasks in the cases that training OOD data are available or not.


**Questions:**

1.In Section 4.2, the authors propose that Theorem 1 explains the independence assumption is beneficial for OOD detection.  Is there any clue to prove that conditional independence within a class is more effective than restricting independence between inliers and outliers? Are there any empirical experiments to support this idea?

2.The impact of using different augmentation methods on the result should be explored in the ablation. In Image OOD detection tasks, why use two different methods of generating fake outlier samples? Why not generate them in a uniform way?

3.How about the HOOD results on the unseen OOD training data setting?

4.How much computational cost can be saved by using the memory bank architecture?


**Limitations:**

The authors seemed not to  discuss the limitations of the proposed model.

**Strengths And Weaknesses:**

Strengths:

The work relies on an explicit independence hypothesis, i.e., the inliers  extract only little predictive information from outliers during training. The authors extend the assumption to the conditional independence which encourages each IND class to produce independent features from outliers. Moreover, theoretical arguments and supporting analyses are provided for this independence assumption. The theoretical derivations basically follow the work from [2] and [38], which seem to be correct and convincing to me.

Weaknesses:

1.The innovation of the article seems limited to me,  mainly since the work shares the same perspective as [2]. Both the models build upon the probabilistic formulation and applies the Hilbert-Schmidt Independence Criteria (HSIC).  It may be good to clarify a bit more on how novel the paper is compared from [2].

2.There is a lack of qualitative experiments to demonstrate the validity of the conditional independence model.
a)It is better to provide some  illustrative experimental results to demonstrate that minimising HSICcond-i could indeed perform better than minimising HSIC_HOOD. Possibly, one toy dataset can be used to demonstrate the separability of inlier features and outlier features.

b)The authors propose a new test metric, however, lacking the correctness test and comparative experiments with other metrices.
It may be better to provide some visualization results or schematic diagram, which could  make  readers easier to understand.

3.Current experimental results seem not very convincing to me. Some critical comparative results are missing.
a)Under the setting of unseen OOD training data, DIN [34], Mahalanobis distance [31], Energy [36], their original papers did not use fake/augmented OOD training data. These settings need to be clarified in the paper. Moreover, the impact of using different augmentation methods on the result could  be explored in the ablation.

b)In CIFAR-100, the experimental setup appears to be consistent with that of HOOD [2]. However, in Table 1 (unseen OOD training data), the HOOD’s results are missing.  In [2], the results of HOOD are superior to those of Conditional-I's.

c)In Table 2 (unseen OOD training data), the HOOD’s results are also missing.

d)There are missing both Conditional-i-generative and HOOD results for the NLP OOD detection tasks.
As missing the results of the most relevant methods,  the present experiments could not  convince me of  the validity of the improvements.

4.The memory bank architecture is one contribution, but the authors do not provide quantitative results of introducing the memory bank architecture.

---

> ### Author Response · Authors · 2022-08-02
> **Response to Reviewer 62WK (Part 5/5)**
>
> (continued from Part 4/5)
>
> **Q2**: The impact of using different augmentation methods on the result should be explored in the ablation. In Image OOD detection tasks, why use two different methods of generating fake outlier samples? Why not generate them in a uniform way?
>
> **Ans-Q2**: We used two different augmentations, with the motivation to verify the flexibility and robustness of Conditional-i on different augmentation methods. However, it is a good suggestion to also ablate out the effectiveness of the augmentation types.
>
>
> During the rebuttal time, we were able to re-train the ImageNet1K experiment (Table 2 in the main paper) using different augmentations. We use "+Cutmix" to indicate the Cutmix augmentation whereas we use "+RandAug" to denote the strong augmentation.  The ablation study is shown as below (Other training details are the same as Table 2 in main paper):
>
> | **Methods**         | **FPR95**     | **AUROC**     | **AUPR**      |
> |-----------------|-----------|-----------|-----------|
> | K-base          | 66.05     | 72.98     | 32.93     |
> | +Cutmix         | 61.45     | 75.77     | 42.17     |
> | +RandAug        | 56.15     | 79.31     | 42.47     |
> | +RandAug+Cutmix | **50.20** | **81.20** | **44.55** |
>
> As can be seen from the table above, strong augmentation along with the Cutmix (i.e., +RandAug+Cutmix) would lead to the best results, whereas other augmentation methods would also help Conditional-i to beat SOTA methods such as ReACT and Energy.
>
> **Q3**: How about the HOOD results on the unseen OOD training data setting?
>
> **Ans-Q3**: Please see our reply in Ans-W3 b), Ans-W3 c), Ans-W3 d).
>
> **Q4**: How much computational cost can be saved by using the memory bank architecture?
>
> **Ans-Q4**: If there were no memory bank architecture, we would have to compute the Eq.(9) through batchwise training only, i.e., each in-distribution training batch needs to dynamically maintain sampled $\hat C$ classes, while each class has N samples, i.e., $N\times \hat C$ samples. Such required huge batchsize would quickly drain GPU memory and turns out to be a great challenge for practical training of Conditional-i.

---

> > ### Comment · Reviewer_62WK · 2022-08-07
> > **Significance of conditional-i**
> >
> > The reviewer appreciated the great efforts from the authors on the further quantitative experiments, which resolved some of the previous concerns.
> >
> > Overall,  conditional-i seems to only work with fake OOD training data in the case  without access to real OOD data. The theoretical explanation clarifies the difference between conditional-i and HOOD, and quantitative results demonstrate conditional-i appears better.  However,  I  am not yet convinced  by the significance of conditional-i: why sample-based is significantly superior to  class-based, especially as  sample-based seems less efficient due to  its reliance on memory bank.   As such,  I am not very  comfortable if  the novelty and/or significance of conditional-i  can be deemed as "enough" or just "incremental"  compared with HOOD, though they are indeed different. It would be the best if some qualitative experiments/examples/visualisations/iilustrations could be given showing the distinct superiority of conditional-i over HOOD. Namely, it would be good if the authors may identify/show some scenarios where conditional-i works but HOOD may not work.

---

> > > ### Author Response · Authors · 2022-08-08
> > > **Response to Reviewer 62WK Round 2 (Part 2/2)**
> > >
> > > **(Continued from part 1/2, round 2)**
> > >
> > > In terms of concrete examples, we provided a concrete example in the previous response, where under the scenario $d=1$,  conditional-i works but HOOD fails. But let's see more examples here then. We use $HSIC_{hood}$ to represent the HOOD HSIC loss, and we use $ HSIC_{cond-i} $ to represent the HSIC loss of  Conditional-i.
> > >
> > >
> > > - **Example 1 where HOOD fails and Conditional-i works: $d=1$**
> > >
> > > Here we provide an example to demonstrate the difference in solution between the Conditional-i and HOOD methods.
> > > With results of Sections 3.4 and 4.1 in the main text, consider in the scenario of $d=1$ and linear kernel, $HSIC_{hood} = \vert {\boldsymbol{Z}}^{\top}\boldsymbol{Q} \vert_F$, where $\boldsymbol{Z}$ denotes the vector of samples from the mixture distribution $p_M(\boldsymbol{z}) = \sum_{c=1}^C p_\theta(\boldsymbol{z}^{(\xi)}\mid \xi = c) p(\xi = c)$ and $\boldsymbol{Q}$ denotes the samples of out of distribution features $\boldsymbol{q}$. When the sample size is large, $HSIC_{hood}$ is approximately the Frobenius norm of $N$ times of the correlation between the out of distribution feature and $\boldsymbol{z}$ from the mixture distribution.
> > > Minimizers $\boldsymbol{q}$ of $HSIC_{hood}$ are those orthogonal (inner product defined by the correlation) to the linear combination $\sum_{c= 1}^C p(\xi = c) \boldsymbol{z}^{(c)}$, where $\boldsymbol{z}^{(c)}$ follows the conditional distribution of $\boldsymbol{z}$ given class $\xi = c$.
> > > Suppose the total number of classes $C=2$, $p(\xi = 1) = p(\xi = 2) = 1/2$, and the correlation between $z^{(1)}$ and $z^{(2)}$ is omissible.
> > > Then $\boldsymbol{q} = \boldsymbol{z}^{(1)}/\sqrt{2} - \boldsymbol{z}^{(2)}/\sqrt{2}$ minimizes $HSIC_{hood}$ but is not orthogonal of each $\boldsymbol{z}^{(c)}$.
> > > On the contrary, the minimizer of $HSIC_{cond-i} $ is required to be orthogonal of each $\boldsymbol{z}_i^{(c)}$.
> > >
> > > - **Example 2 where HOOD fails and Conditional-i works: $d=3$**
> > >
> > > In the scenario of $d=3$ and linear kernel, $HSIC_{hood} = \vert {\boldsymbol{Z}}^{\top}\boldsymbol{Q} \vert_F$. We assume, for the purpose of visual illustration, in-distribution feature from the 1st in-distribution class ${\boldsymbol{z}}^{(1)}$ is virtually represented by the vector $[1,0,0]^{\top} \in R^{3}$, and the 2nd in-distribution class ${\boldsymbol{z}}^{(2)}$ is virtually represented by $[0,1,0]^{\top} \in R^{3}$. Suppose again the total number of classes $C=2$, $p(\xi = 1) = p(\xi = 2) = 1/2$, and the correlation between $z^{(1)}$ and $z^{(2)}$ is omissible, made apparent by the orthogonal relation between $z^{(1)}$ and $z^{(2)}$. Then OOD feature $\boldsymbol{q} = [-\frac{\sqrt{2}}{2}, \frac{\sqrt{2}}{2}, 0 ]$ is one of the minimizers for $HSIC_{hood}$ but is not orthogonal of each individual $\boldsymbol{z}^{(1)}$ or $\boldsymbol{z}^{(2)}$ from in-distribution classes at all. This means that $\boldsymbol{q}$ still poses strong correlations/dependence with both in-distribution features ${\boldsymbol{z}}^{(1)}$ and ${\boldsymbol{z}}^{(2)}$, whereas $\boldsymbol{q}$ is independent of the standardized sum $[\frac{\sqrt{2}}{2}, \frac{\sqrt{2}}{2}, 0 ]$ of the two  (HOOD motivation), which represents the mixture distribution of the two features.
> > > However, the HOOD minimizer $\boldsymbol{q} = [-\frac{\sqrt{2}}{2}, \frac{\sqrt{2}}{2}, 0 ]$ cannot be a minimizer of $ HSIC_{cond-i} $, as Conditional-i poses stricter conditions that $\boldsymbol{q}$ needs to be having small dependence on every in-distribution class. Correspondingly, $\boldsymbol{q} = [0, 0, 1 ]$ is an optimal solution of Conditional-i loss, that $\boldsymbol{q} = [0, 0, 1]$ achieves simultaneous independence with both ${\boldsymbol{z}}^{(1)}$ and ${\boldsymbol{z}}^{(2)}$. Note that the desired solution $\boldsymbol{q} = [0, 0, 1]$ can also be a local minimum of HOOD though, but HOOD has no incentive to avoid the solution of $\boldsymbol{q} = [-\frac{\sqrt{2}}{2}, \frac{\sqrt{2}}{2}, 0 ]$ among infinite number of local minima, whereas Conditional-i exclusively can escape such bad local minima, by always anchoring at the desired solution $\boldsymbol{q} = [0, 0, 1]$. **Please also refer to Section 10 in our supplementary file revision for illustration on this example.**
> > >
> > > - **More Visualizations in the supplementary file revision**
> > >
> > > We also associate new visualization results to support the difference between HOOD and Conditional-i. Please view our supplementary material revision for these new visualizations. In supplementary file, Section 10 demonstrates how well the test score functions out of each method separates the in-distribution and out-of-distribution data. This well illustrates that Conditional-i offers better separation than HOOD, as per the discussion above.
> > >
> > > We sincerely hope these evidence may reassure Reviewer 62WK that HOOD and Conditional-i indeed correspond to different optimization models. Conditional-i offers both exclusive theoretical and empirical superiority over HOOD.

---

> > > > ### Comment · Reviewer_62WK · 2022-08-09
> > > > **Thanks for the response**
> > > >
> > > > Thank the authors for the nice response.  I am now convinced that conditional-i can be very different from HOOD as evidenced by the additional examples and visualisations. I am happy to raise my rating accordingly.

---

> > > > > ### Author Response · Authors · 2022-08-09
> > > > > **Thanks for your time!**
> > > > >
> > > > > Dear Reviewer 62WK,
> > > > >
> > > > > We are glad that our answers were helpful! We really appreciate your kind engagement during this rebuttal process.
> > > > >
> > > > > Best Regards

---

> > > ### Author Response · Authors · 2022-08-08
> > > **Response to Reviewer 62WK Round 2 (Part 1/2)**
> > >
> > > We are very grateful for the reviewer's stimulating comments and questions, which encouraged us to ponder more on the theoretical advantages of Conditional-i comparing to HOOD as well as the essential difference between the two approaches.
> > >
> > > Further thoughts on theoretical justification of Conditional-i comparing to HOOD are presented here. For simplicity we first consider the case of infinite sample size $N=\infty$, which represents the ideal model. Under this scenario, the optimal solution to Conditional-i enjoys the perfect feature extraction of the OOD samples and the perfect disentanglement from in-distribution generators in the sense that the extracted features $\boldsymbol{q}$ is an invertible function of the true OOD generator $\beta$ while not being influenced by the in-distribution generator $\omega$. This can be shown by considering the Jacobian of $\boldsymbol{q}$ as a function of $\omega$, and using the Conditional-i constraints of $\boldsymbol{q}$ independent of each in-distribution class's feature $\boldsymbol{z}^{(c)}$ in the chain rule of calculating the derivatives. HOOD, on the contrary, does not possess this good property, as the independence constraint is only on the mixture distribution of all classes' features as opposed to on each individual class. In the case of finite sample size $N < \infty$, we no longer have the perfect disentanglement, but a similar bound for $\frac{\partial \boldsymbol{q}}{\partial \omega}$ can be derived with the approach similar to that of Theorem 1 for the Conditional-i approach. Note this new bound will be a ``mirror'' result to that of Theorem 1. Theorem 1 describes the disentanglement of in-distribution features from OOD generators, and the new result describes the disentanglement of OOD features from in-distribution generators.
> > >
> > > (**To be continued in part 2/2, Round 2**)

---

> ### Author Response · Authors · 2022-08-02
> **Response to Reviewer 62WK (Part 4/5)**
>
> (continued from Part 3/5)
>
> **W4**: The memory bank architecture is one contribution, but the authors do not provide quantitative results of introducing the memory bank architecture.
>
> **Ans-W4**: We are unsure if we understand the request here, but the ablation study on the memory bank architecture was already explicitly presented in the paper, see Section 5.4 ablation study, paragraph "Ablation on memory bank architecture" (line 349-363). Since the class-wise (Conditional) independence was a nuisance of the training implementation, the memory bank architecture is introduced in order to circumvent this difficulty by efficiently storing all the training classes. The ablation experiments in this part well justified that, introducing the memory bank efficiently eases the implementation of Conditional-i, whereas applying the same bank architecture to HOOD may hardly benefit from that architecture. This highlights the importance of the Conditional-i generative assumption (instead of the memory bank architecture itself), by virtue of the analysis in Section 4.1 and in Theorem 1.
>
> **Q1**: In Section 4.2, the authors propose that Theorem 1 explains the independence assumption is beneficial for OOD detection. Is there any clue to prove that conditional independence within a class is more effective than restricting independence between inliers and outliers? Are there any empirical experiments to support this idea?
>
> **Ans-Q1**: Please also see Ans-W1 that responds to W1 on similar concerns. Note that our independence constraints Eq. (11) are crucial in disentangling the extracted OOD features from the in-distribution features and generators. Constraining on the independence between the OOD and each in-distribution class's features guarantees the OOD feature extractor is an invertible function of the OOD generator alone without mixing with in-distribution generators.
>
> Here we provide an example to demonstrate the difference in solution between the Conditional-i and HOOD methods.
> With results of Sections 3.4 and 4.1 in the main text, consider in the scenario of $d=1$ and linear kernel, $HSIC_{hood} = \vert {\boldsymbol{Z}}^{\top}\boldsymbol{Q} \vert_F$, where $\boldsymbol{Z}$ denotes the vector of samples from the mixture distribution $p_M(\boldsymbol{z}) = \sum_{c=1}^C p_\theta(\boldsymbol{z}^{(\xi)}\mid \xi = c) p(\xi = c)$ and $\boldsymbol{Q}$ denotes the samples of out of distribution features $\boldsymbol{q}$. When the sample size is large, $HSIC_{hood}$ is approximately the Frobenius norm of $N$ times of the correlation between the out of distribution feature and $\boldsymbol{z}$ from the mixture distribution.
> Minimizers $\boldsymbol{q}$ of $HSIC_{hood}$ are those orthogonal (inner product defined by the correlation) to the linear combination $\sum_{c= 1}^C p(\xi = c) \boldsymbol{z}^{(c)}$, where $\boldsymbol{z}^{(c)}$ follows the conditional distribution of $\boldsymbol{z}$ given class $\xi = c$.
> Suppose the total number of classes $C=2$, $p(\xi = 1) = p(\xi = 2) = 1/2$, and the correlation between $z^{(1)}$ and $z^{(2)}$ is omissible.
> Then $\boldsymbol{q} = \boldsymbol{z}^{(1)}/\sqrt{2} - \boldsymbol{z}^{(2)}/\sqrt{2}$ minimizes $HSIC_{hood}$ but is not orthogonal of each $\boldsymbol{z}^{(c)}$.
> On the contrary, the minimizer of $HSIC_{cond-i} $ is required to be orthogonal of each $\boldsymbol{z}_i^{(c)}$.
>
> This is a perfect example demonstrating the capability of the proposed Conditional-i method in disentangling out-of-distribution features from the in distribution features and generators, where HOOD fails.
>
> (To be continued in **Part 5/5**).

---

> ### Author Response · Authors · 2022-08-02
> **Response to Reviewer 62WK (Part 3/5)**
>
> (continued from Part 2/5)
>
> **W3 a)**: Current experimental results seem not very convincing to me. Some critical comparative results are missing. a)Under the setting of unseen OOD training data, ODIN [34], Mahalanobis distance [31], Energy [36], their original papers did not use fake/augmented OOD training data. These settings need to be clarified in the paper. Moreover, the impact of using different augmentation methods on the result could be explored in the ablation.
>
> **Ans-W3 a)**: Frankly speaking, it seems a bit unfair to blame Conditional-i for seeing synthetic data. Instead of viewing it as an unfairness, we would rather regard it as the exclusive advantage of Conditional-i to leverage on such synthesized OOD training data, whereas other counterparts test method such as ODIN, ReACT, Mahalanobis-distance, were inherently unable to exploit these useful information. Since it is non-trivial to adapt these existing test methods to benefit from the fake OOD data, we kindly think that it shall not be on our burden to innovate these methods.
>
> But admittedly, it is still a good idea to train OE and Energy with the generated OOD data, in order to justify the advantage Conditional-i and HOOD. Here we use the same augmentation setting (+RandAug+Cutmix) as described in paper for producing Table 2 in main paper (see supplementary file Section 4 line 56-73), and we train OE and Energy with the fake OOD data on IN1K. The results are presented as follows:
>
>
> | **Methods**        | **FPR95**     | **AUROC**     | **AUPR**      |
> |----------------|-----------|-----------|-----------|
> | K-base         | 66.05     | 72.98     | 32.93     |
> | OE-generative            | 63.43     | 74.11     | 38.13     |
> | Energy-generative         | 54.51     | 79.54     | 43.59     |
> | HOOD-generative           | 52.28     | 80.16 | 44.38     |
> | Conditional-i-generative  | **50.20** | **81.20**     | **44.55** |
>
>
> The above results well demonstrate that Conditional-i outperforms all its counterparts that were exposed to fake OOD training data, strongly echoing our claims in the paper.
>
> **W3 b)**: In CIFAR-100, the experimental setup appears to be consistent with that of HOOD [2]. However, in Table 1 (unseen OOD training data), the HOOD’s results are missing. In [2], the results of HOOD are superior to those of Conditional-i's.
>
> **Ans-W3 b)**: Actually, the backbones for training the CIFAR100 experiments in the two papers are different. If we look at HOOD paper [2] by referring to "Section 5.3 Training setup", one would notice we used WideResNet-40-2 for CIFAR 100 training in [2], whereas in this NeurIPS paper (Line 303, Section 5.2 Training architecture and optimization), we instead used WideResNet-28-10 for training. This change of architecture has introduced performance gap between the two papers.
>
> **W3 c)**: In Table 2 (unseen OOD training data), the HOOD’s results are also missing.
>
> **Ans-W3 c)**: We have displayed the results in the Table in **Ans-W3 a)**, where Conditional-i always provides superior performance than other baseline methods.
>
> **W3 d)**: There are missing both Conditional-i-generative and HOOD results for the NLP OOD detection tasks. As missing the results of the most relevant methods, the presented experiments could not convince me of the validity of the improvements.
>
> **Ans-W3 d)**: It is a legitimate question asking for NLP result of Conditional-generative. By following the similar logic how we trained for CV tasks, we adopt the strong augmentation technique in NLP paper [R2A] so that we can generate fake OOD training data for NLP tasks. We then present the results of Conditional-i , HOOD and OE trained with such fake NLP OOD training data. The results are presented as follows:
>
>
> | **Methods**                  | **FPR95**     | **AUROC**     | **AUPR**      |
> |--------------------------|-----------|-----------|-----------|
> | K-base                   | 52.84     | 84.27     | 41.41     |
> | OE-generative            | 49.89     | 86.20     | 61.95     |
> | Hood-generative          | 37.92     | 92.51     | 78.66     |
> | Conditional-i-generative | **33.17** | **93.12** | **78.94** |
>
> Note: EDA contains Synonym Replacement, Random Insertion, Swap, and Deletion.
> [R2A] J. Wei, K. Zou, ''EDA: Easy Data Augmentation Techniques for Boosting Performance on Text Classification Tasks'', EMNLP-IJCNLP 2019
>
> This again verifies the benefit of Conditional-i over all the counterpart baselines on NLP tasks.
>
> (To be continued in **Part 4/5**).

---

> ### Author Response · Authors · 2022-08-02
> **Response to Reviewer 62WK (Part 2/5)**
>
> (continued from Part 1/5)
>
> **W2**: There is a lack of qualitative experiments to demonstrate the validity of the conditional independence model.
> W2 a): It is better to provide some illustrative experimental results to demonstrate that minimising HSICcond-i could indeed perform better than minimising HSIC_HOOD. Possibly, one toy dataset can be used to demonstrate the separability of inlier features and outlier features.
>
> **Ans-W2 a)**: Before presenting any toy data based experiments, we kindly ask Reviewer to re-evaluate the soundness of our paper based on the existing experiments on real data. We follow the protocols defined in OE, energy, by training on 3 different benchmarks and test on in total 15 test sets including both computer vision and NLP tasks, all of which showing evident empirical gain over HOOD. It is frustrating though, to see that our paper is deemed "lack of qualitative experiments to demonstrate the validity of the conditional independence model". We also kindly encourage the Reviewer 62WK to take into account the other supplementary results we provide in this rebuttal, which demonstrate evident superiority of Conditional-i over HOOD and other counterparts under other challenging scenarios and more detailed ablations.
>
> We are not quite sure though, what toy data is required here to further demonstrate the merits of Conditional-i better beyond real data. Perhaps we are unable to provide new toy data experiment at this tight time during this short rebuttal though. However, we can provide other quantitative evidence that Conditional-i indeed can reduce the HSIC metric among all the counterparts. Since the HSIC metric is guaranteed to reflect the mutual information, the Table below effectively proves that our method to the most extent can remove the spurious correlation between in and OOD data during training, thus better discriminating the OOD data during test. Specifically, we employ Eq. (5) to compute the HSIC metric of the deep models out of every compared method at the last training epoch (under the same training setup as we produce Table 1). We then compute the HSIC values of the compared methods using CIFAR-100 test data and OOD test data respectively as $\Phi_z^{(c)}$ and $\Phi_q$. Conditional-i returns the smallest HSIC loss which is reflective of its corresponding small mutual information between in- and out-distribution features.
>
> | **Metrics Values** | **HSIC loss Gaussian kernel ($\tau=5$)** | **HSIC loss linear kernel** | **FPR95** $\downarrow$ | **AUROC** $\uparrow$ | **AUPR** $\uparrow$ |
> | ------------------ | ---------------------------------------- | ---------------------------- | ---------------------- | --------------------- | -------------------- |
> |   K-Base              | 0.0053       |          16.2912                |        65.84           |    74.76     |  33.25 |
> |         OE               |    0.0019    |        4.2954               |        33.41         |     89.75     |  57.74 |
> |         Conditional-i     | 0.0003      |             0.4850       |            32.66       |      90.03    |  61.48 |
>
> **W2 b)**: The authors propose a new test metric, however, lacking the correctness test and comparative experiments with other metrics. It may be better to provide some visualization results or schematic diagram, which could make readers easier to understand.
>
> **Ans-W2 b)**: Like Energy and OE, Our conditional-i is a holistic procedure, too. Note that our test Eq. (9) was originally motivated from the training objective Eq. (7) which exclusively couples with our Conditional-i training loss. Nevertheless, it still worths to try implementing the Conditional-i test/train alone independently. In the Table below, we use Train (Test) to represent the model trained with "Train" method and tested with "Test" method. We apply the Conditional-i test on the model trained via conventional cross entropy loss, which is denoted as K-base (Cond-i). In contrast to the model K-base (SFM) which employs a softmax score (SFM) for test, we see K-base (cross-entropy) trained model may hardly benefit from the new test. In the meanwhile, we also observe that Conditional-i (SFM) may hardly beat Conditional-i (Cond-i), showing the critical role of coupled train/test pair. This again verifies that Conditional-i is a holistic procedure with principled motivation and theoretical support.
>
> | **Methods**                | **FPR95**     | **AUROC**     | **AUPR**      |
> |------------------------|-----------|-----------|-----------|
> | K-base (SFM)           | 66.05     | 72.98     | 32.93     |
> | K-base (Cond-i)        | 70.69     | 69.31     | 37.48     |
> | Conditional-i (SFM)    | 56.76     | 81.20     | 48.37     |
> | Conditional-i (Cond-i) | **42.84** | **86.79** | **55.00** |
>
> Regarding the diagram, we already included an algorithmic flow in the supplementary material, see "Section 5 Overall Algorithm Flow of Conditional-i" in supplementary file. (To be continued in **Part 3/5**).

---

> ### Author Response · Authors · 2022-08-02
> **Response to Reviewer 62WK (Part 1/5)**
>
> We appreciate the constructive comments from Reviewer 62WK and the time Reviewer 62WK spent on reviewing the paper. We also thank Reviewer 62WK's kind patience in engaging in this rebuttal. Let us address some key confusions and concerns here. We hope the information here may further supplement our main submission and justify the soundness of Conditional-i. We will include the primary results and discussion here in the main paper.
>
> **W1**: The innovation of the article seems limited to me, mainly since the work shares the same perspective as [2]. Both the models build upon the probabilistic formulation and applies the Hilbert-Schmidt Independence Criteria (HSIC). It may be good to clarify a bit more on how novel the paper is compared from [2].
>
> **Ans-W1**: We would like to emphasize that paper [2] was an earlier trial that conveys very preliminary trails and limited results that were submitted earlier to another venue (under review), which in certain cases HOOD [2] would probably fail in discriminating OOD samples. In this regard, please also refer to analysis in Section 4.1 in our Conditional-i paper, and also see supplementary material line 43-55. We were happy to share that earlier idea [2] with the community, whereas we think Conditional-i should not to be criticised for continuously challenging the usefulness of the HSIC hypothesis.
>
>
> In fact, the Conditional-i paper points out many limitation of HOOD, while we associate rigorous proofs to support these claims. We also encourage the reviewer to re-consider the details given in Section 4.1, which also manifests itself that the optima of HOOD and Conditional-i would be drastically different. Our independence constraints Eq. (11) are crucial in disentangling the extracted OOD features from the in-distribution features and generators. Constraining on the independence between the OOD and each in-distribution class's features guarantees the OOD feature extractor is an invertible function of the OOD generator alone without mixing with in-distribution generators.
>
> By virtue of the analysis above, Conditional-i shows strong empirical superiority over HOOD, by successfully (and theoretically) circumventing these limitations and failure cases (through construction of novel memory bank structures). Note that this paper also considers using a brand new generative process hypothesis that were especially tailored for Conditional-i, which also supports the construction of Theorem 1.  Such specific generative modeling does not transfer to HOOD [2] owing to its special assumption set, and HOOD also did not enjoy similar guarantee bounds of Conditional-i. Perhaps the last but not least, the test of conditional-i is distinct from that used in HOOD in principle. HOOD [2] introduces a simple linear correlation test that measures between the features that does not sufficiently reflect the independence. However, the new test here was particularly motivated to better reflect the independence metric in a more principled way.
>
> Please also see Ans-Q1 that responds to Q1 on similar concerns. We sincerely appreciate it if Reviewer may kindly reconsider the above points during this rebuttal.
>
> (To be continued in **Part 2/5**).

---

### Official Review · Reviewer_FRYP · 2022-07-11

**Rating:** 6
**Confidence:** 3
**Soundness:** 3 good
**Presentation:** 2 fair
**Contribution:** 3 good

**Summary:**

This paper proposes a new out-of-distribution (OOD) detection method that makes use of known OOD data. Given a classifier with an encoder that outputs features, and a linear classification layer, the idea is to decorrelate the features of in-distribution and OOD samples in a class-wise manner. In order to achieve this, the authors use the Hilbert-Schmidt Independence Criterion (HSIC) to measure the independence between in-distribution (of a given class) and OOD features. A classifier is then trained on the usual classification loss plus the independence objective. However, since computing the HSIC metric involves computing the features of $N$ in-distribution samples in every class, which can get expensive with large $N$, the authors use a buffer to reuse old features and save compute. The method, which is called Conditional-i, is evaluated on OOD detection benchmarks in image classification and NLP, and achieves better performance than other baseline methods.

**Questions:**

- As mentioned previously, how were the values of $\lambda$ and $\tau$ decided? Was a separate validation OOD dataset used?
- How reasonable are the assumptions for Theorem 1, especially the ones involving $g$’s?
- The authors mention in lines 267-268 that “This hopefully may better generalize the training to unseen test OODs as long as the generative process of the test OOD is genuinely different”. What does a different generative process mean, and how can we define it? I’m not sure I believe this statement since Theorem 1 doesn’t guarantee anything about the unknown OOD case.
- In the experiments, have the authors tried using just the test scoring method from Equation 9 on a classifier trained normally with cross entropy?
- It doesn’t seem fair comparing Conditional-i-generative with methods that don’t use any auxiliary data (even if it’s synthetic). Have the authors tried outlier exposure with the generated OOD data?
- Have the authors removed overlapping images between train in-distribution, train OOD and test OOD? For example, the tiny images dataset contain CIFAR-10 and CIFAR-100.
- What $\hat{C}$, $N$ values were used for each experiment?
- What were the test classification accuracies achieved by Conditional-i on the in-distribution dataset?

Other comments:
- Line 204: “matrics” -> “metrics”
- Line 218: Typo in $p_M(\mathbf{z}) = \sum_{c=1}^C p_\theta(\mathbf{z}^{(\xi)},\mathbf{q}|\xi=c)p(\xi=c)$ -> $p_M(\mathbf{z}) = \sum_{c=1}^C p_\theta(\mathbf{z}^{(\xi)}|\xi=c)p(\xi=c)$
- Table 1 and 2 caption: “trainnig” -> “training”
- It would be nice to include the setup that was used to compute the values in Table 4, 5, and Figure 1.


**Limitations:**

The authors used the Tiny Images dataset, which was withdrawn for ethical reasons.


**Strengths And Weaknesses:**

Strengths:
- The proposed method is very similar to HOOD, as the authors mentioned, however, the differences (class-wise computation of HSIC, and usage of memory bank), are made clear, and there is evidence of improvement in the experiments.
- The proposed method seems to consistently perform better than other OOD detection methods.

Weaknesses:
- There is no guarantee that decorrelating the features of in-distribution and known OOD data will lead to improved feature distinction between in-distribution and unknown OOD data.
- There is no mention in the paper how hyperparameters ($\lambda$ and $\tau$) were tuned for each of the methods. However, it seems from the ablation experiments that the used values conveniently match up with the best performance on the test OOD datasets. If it is true that the hyperparameters were tuned on the test set, the results presented in the paper aren’t trustworthy.

---

> ### Author Response · Authors · 2022-08-02
> **Response to Reviewer FRYP (Part 3/3)**
>
> (continued from Part 2/3)
>
> **Q5**: It doesn’t seem fair comparing Conditional-i-generative with methods that don’t use any auxiliary data (even if it’s synthetic). Have the authors tried outlier exposure with the generated OOD data?
>
> **Ans-Q5**:  Frankly speaking, it seems a bit unfair to blame Conditional-i for seeing synthetic data. Instead of viewing it as an unfairness, we would rather regard it as the exclusive advantage of Conditional-i to leverage on such synthesized OOD training data, whereas other counterparts test method such as ODIN, ReACT, Mahalanobis-distance, were inherently unable to exploit these useful information. Since it is non-trivial to adapt these existing test methods to benefit from the fake OOD data, we kindly think that it shall not be on our burden to innovate these methods.
>
> But admittedly, it is still a good idea to train OE and Energy with the generated OOD data, in order to justify the advantage Conditional-i and HOOD. Here we use the same augmentation setting (+RandAug+Cutmix) as described in paper for producing Table 2 in main paper (see supplementary file Section 4 line 56-73), and train OE and Energy with the fake OOD data on IN1K. The results of the compared methods are:
>
> | **Methods**        | **FPR95**     | **AUROC**     | **AUPR**      |
> |----------------|-----------|-----------|-----------|
> | K-base         | 66.05     | 72.98     | 32.93     |
> | OE-generative             | 63.43     | 74.11     | 38.13     |
> | Energy-generative         | 54.51     | 79.54     | 43.59     |
> | HOOD-generative           | 52.28     | 80.16 | 44.38     |
> | Conditional-i-generative  | **50.20** | **81.20** | **44.55** |
>
> Again, this shows evident performance gain of Conditional-i-generative in comparison to all the baselines.
>
> **Q6**: Have the authors removed overlapping images between train in-distribution, train OOD and test OOD? For example, the tiny images dataset contain CIFAR-10 and CIFAR-100.
>
> **Ans-Q6**: Yes, we strictly follow the training protocol defined in OE paper [26], and we carefully made sure to exclude the CIFAR-10 and CIFAR-100 dataset. This description can also be found in OE paper [26], in "Section 4.2.2 Outlier Exposure Datasets". It reads "we remove all examples of 80Million Tiny Images which appear in the CIFAR datasets..."
>
> **Q7**: What $\hat C$ and $N$ values were used for each experiment?
>
> **Ans-Q7**: The $\hat C$ and $N$ values in Table 1,2,3, were respectively defined as $\hat C=100, 1000, 20$, $N=128, 256, 64$.
>
> **Q8**: What were the test classification accuracies achieved by Conditional-i on the in-distribution dataset?
>
> **Ans-Q8**:  We illustrate the corresponding top-1 acc on the in-distribution data (for Table 1,2,3 in the main paper) as follows:
>
> | **Methods**        | **top-1 (CIFAR-100)** | **top-1 (IN1K)** | **top-1 (20NG)** |
> |----------------|-------------------|--------------|--------------|
> | K-base         | **80.76**         | 54.01        | 26.31        |
> | OE             | 78.33             | **53.55**    | 28.21        |
> | Energy         | 77.72             | 52.14        | 27.91        |
> | HOOD           | 77.83             | 53.10        | 28.30        |
> | Conditional-i  | 77.21             | 52.87        | **28.36**    |
>
> As can be seen from the above table, we indeed occasionally sacrifice a bit classification accuracy on the in-distribution data. However, it seems this is a worthwhile trade-off, since Conditional-i can effectively boost the OOD detection performance to a large extent at the merely marginal cost of classification accuracy.
>
> **Limitations:** The authors used the Tiny Images dataset, which was withdrawn for ethical reasons.
>
> **Ans:** We followed the OE and Energy to set up the training procedure and training data so that we can compare with these SOTAs. We apologize for not realizing this issue earlier. We shall be able to supplement the paper by instead using ImageNet22K data and update the results.

---

> > ### Comment · Reviewer_FRYP · 2022-08-08
> > **Response**
> >
> > I want to thank the authors for their detailed response to all the reviews.
> >
> > **Response to Ans-W1**
> >
> > What exactly is $\beta$? Is it just a variable that is used to prove the theorem? Or is it explicitly modeled in the algorithm? It seems to me that $\beta$ has to be different for different possible OOD classes. For example, a distribution over textures should have a different $\beta$ than a distribution over CIFAR100. Please let me know if I am misunderstanding something.
> >
> > **Ablation study name change**
> >
> > I forgot to include it in my main review, but I don’t think it’s appropriate to use “ablation study” for the experiments where the value of parameters are varied (experiments varying $\hat{C}$, $N$ and $\lambda$).
> >
> > My other questions and concerns were addressed by the authors’ response. I have read all the other reviews, and so far I am convinced that the work is novel compared to HOOD, and it works in practice based on the experiments. Therefore, I increased my score to accept this paper.

---

> > > ### Author Response · Authors · 2022-08-08
> > > **Thanks for further comments**
> > >
> > > We are glad that our response has clarified your concerns! We also feel very grateful to the many constructive comments from Reviewer FRYP.
> > >
> > > Your question is actually very intriguing. In fact, $\beta$ is the generator of a generic OOD sample, and was defined for theoretical purpose. If there are multiple OOD types, then $\beta$ can refer to the generalized generator with a mixture distribution of all possible OOD types, leaving our Theorem 1 unchanged.
> > >
> > > We included the experiments on $\lambda$, $N$ and $\hat C$ in ablation study so that we can demonstrate the effect of different hyperparameters. However, we can certainly modify the section header so that this part of experiments are named "sensitivity against hyperparameters" or the likes.
> > >
> > > Thanks again for your time!

---

> ### Author Response · Authors · 2022-08-02
> **Response to Reviewer FRYP (Part 2/3)**
>
> (continued from Part 1/3)
>
> **Q1**: As mentioned previously, how were the values of $\lambda$ and $\tau$ decided? Was a separate validation OOD dataset used?
>
> **Ans-Q1**: Yes, we used a separate validation OOD dataset. Please see the above reply "Ans-W2" to Reviewer FRYP for the response to the same question.
>
> **Q2**: How reasonable are the assumptions for Theorem 1, especially the ones involving $g's$.
>
> **Ans-Q2:** The assumptions i)-v) are reasonable and commonly used in literature such as in [38]. Specifically,
>
> - The i) assumption merely assumes that the generative function $g$ and admissible feature extractors $F$ are sufficiently smooth.
> - The ii) assumption constraints the Rademacher complexity so that the class of admissible feature extractors $F$ is not too complex. Central to this assumption is that the feature extractors $F$ can range from a generalized class of functions as opposed to a single function, and the assumption on the Rademacher complexity of the function class describes the trade-off between the expressiveness of the feature extractors and the sample size.  For example, when the extractor class includes complex models with multiple depths of layers, the feature extractors are often more capable of expressing and recovering the data generators. However, the Rademacher complexity also increases, requiring a larger sample size to achieve the same bound in disentangling the in-distribution data features from the OOD generators.
> - iii) also only assumes the smoothness of function $h$.
> - iv)  means the feature extractors should be expressive enough to approximately recover the generators of the data, and the upper bound $\nu$ characterizes how close the feature extractor can approximate the actual values of the generators.
> - v) means that the extracted features and the data generators are bounded, which is a natural assumption, and $C_f$ describes how large the bound is for the features and generators.
>
> **Q3**: The authors mention in lines 267-268 that “This hopefully may better generalize the training to unseen test OODs as long as the generative process of the test OOD is genuinely different”. What does a different generative process mean, and how can we define it? I’m not sure I believe this statement since Theorem 1 doesn’t guarantee anything about the unknown OOD case.
>
> **Ans-Q3**: Please see Ans-W1 above for response on the similar question.
>
> **Q4**: In the experiments, have the authors tried using just the test scoring method from Equation 9 on a classifier trained normally with cross entropy?
>
> **Ans-Q4**: This is an intriguing question. Note that our test Eq. (9) was originally motivated from the training objective Eq. (7) which exclusively couples with our Conditional-i training loss. Nevertheless, it still worths to try implementing the Conditional-i test/train alone independently. In the Table below, we use Train (Test) to represent the model trained with "Train" method and tested with "Test" method. We apply the Conditional-i test on the model trained via conventional cross entropy loss, which is denoted as K-base (Cond-i). In contrast to the model K-base (SFM) which employs a softmax score (SFM) for test, we see K-base (cross-entropy) trained model may hardly benefit from the new test. In the meanwhile, we also observe that Conditional-i (SFM) may hardly beat Conditional-i (Cond-i), showing the critical role of coupled train/test pair. This again verifies that Conditional-i is a holistic procedure with principled motivation and theoretical support.
>
>
> | **Methods**                | **FPR95**     | **AUROC**     | **AUPR**      |
> |------------------------|-----------|-----------|-----------|
> | K-base (SFM)           | 66.05     | 72.98     | 32.93     |
> | K-base (Cond-i)        | 70.69     | 69.31     | 37.48     |
> | Conditional-i (SFM)    | 56.76     | 81.20     | 48.37     |
> | Conditional-i (Cond-i) | **42.84** | **86.79** | **55.00** |
>
> (To be continued in **Part 3/3**).

---

> ### Author Response · Authors · 2022-08-02
> **Response to Reviewer FRYP (Part 1/3)**
>
> We appreciate the constructive comments from Reviewer FRYP and the time spent on reviewing the paper. We also thank Reviewer FRYP's kind patience in engaging in this rebuttal. Let us address some key confusions and concerns here. We hope the information here may further supplement our main submission and justify the soundness of Conditional-i. We will include the primary results and discussion here in the main paper.
>
> **W1**: There is no guarantee that decorrelating the features of in-distribution and known OOD data will lead to improved feature distinction between in-distribution and unknown OOD data.
>
> **Ans-W1**: We notice that this weakness point reflects similar concerns with question Q3, we therefore merge Ans-W1 and Ans-Q3 by responding to both of the questions here.
>
> In fact, this is an intriguing question. The bound derived from Theorem 1 applies to any OOD data *regardless* of the OOD data is known or unknown. The assumptions also manifest in themselves that  "known/unknown" is not required in our generative assumptions or definitions. What Theorem 1 reveals is: if we were determined to optimize Eq. (10)-(12), i.e., to penalize dependence between $\boldsymbol{z}$ and $\boldsymbol{q}$ (no matter what $\boldsymbol{q}$ is, OOD data or not), then we would be guaranteed to disentangle the feature $\boldsymbol{z}$ from the generative process that produces $\boldsymbol{q}$, i.e., the $\beta$ variable, under the assumption that the true generators of the in-distribution and out-of-distribution samples are independent. Therefore, if we encounter any new samples from the same generative process determined by $\beta$ (or variable closed to $\beta$), we would always be able to ensure that $\boldsymbol{z}$ has small dependence on the generative process $\beta$, i.e., the varying $\beta$ would not cause change in $\boldsymbol{z}$. This then guarantees that OOD generative process determined by $\beta$ has small mutual information to in-distribution data. This theorem therefore directly justifies the generalization of our Conditional-i method on unseen new OOD samples (whether it is known or unknown as OOD data) that were generated closed to the generative process determined by $\beta$. However, like any existing training approaches, Conditional-i also has its own generalization bound. If the test sample lies in the region too far from the that generated by $\beta$ variable, the error bound on that sample would expand, by virtue of [R1A].
>
> **Reference:**
>
> [R1A] Ben-David, Shai, et al. "A theory of learning from different domains", 2010.
>
> **W2**: There is no mention in the paper how hyperparameters ( and ) were tuned for each of the methods. However, it seems from the ablation experiments that the used values conveniently match up with the best performance on the test OOD datasets. If it is true that the hyperparameters were tuned on the test set, the results presented in the paper aren’t trustworthy.
>
> **Ans-W2**: We believe every approach in the literature has their own tuning techniques and sensitivity against hyperparameters. Here in the paper, we actually tuned all the baseline models by referring to the validation data defined in OE paper [26], and choose the best hyperparameter based on AUROC. We found the optimal hyperparamters for test and validation set are neglectable for Conditional-i, in terms of AUROC. To justify that this conclusion also holds true for other OOD detection method, we show here that the hyperparameters for OE [2] (also an outlier exposure approach as Conditional-i is) on the same validation set also returns the same optimal hyperparameter as they reported for test data. The sensitivity test against the hyperparmeters we obtained on the CIFAR 100 datasets are respectively ("val" represents the experiments were tuned on validation set) :
>
>
> | **OE**     | **FPR95(val)** | **AUROC(val)** | **AUPR(val)**  |
> |------------|------------|------------|-----------|
> | weight=0.1 | 32.14      | 90.80      | **67.38** |
> | weight=0.5 | 30.88      | **90.83**  | 65.75     |
> | weight=1.0 | **30.53**  | 90.40      | 61.35     |
> | weight=2.0 | 31.97      | 90.07      | 60.23     |
>
> | **Conditional-i** | **FPR95(val)** | **AUROC(val)** | **AUPR(val)** |
> |---------------|------------|------------|-----------|
> | weight=0.006  | 28.85      | 90.96      | 66.47     |
> | weight=0.01   | 29.03      | 90.91      | 64.63     |
> | weight=0.03   | 28.83      | **91.15**  | **66.83** |
> | weight=0.06   | **28.38**  | 90.21      | 59.95     |
>
> (To be continued in **Part 2/3**).

---

### Author Response · Authors · 2022-08-05
**Revision uploaded**

Dear Reviewers,

We sincerely appreciate the constructive comments from everyone, and we enjoyed this process pretty much! In order to reflect the required empirical evidence in the pdf submission, we have uploaded a pdf revision (blue colored highlights), by following the NeurIPS 2022 rules. We understand that we are allowed to submit pdf revisions by 9th August, so please kindly let us know if any additional result is required in the pdf by this date.  We wish we could have fit all the additional discussions and results here into the paper. However, we apologize that owing to the page limit at this time (9 pages), we unfortunately cannot squeeze every result presented in the rebuttal into the main submission pdf now.  However, we will certainly include the complete results/ablations involved in this discussion into the potential camera ready version (10 pages) or into the supplementary file.

Best Regards

---

### Meta-Review · Area_Chair_Lq1e · 2022-08-28

**Recommendation:** Accept
**Confidence:** Certain

**Metareview:**

This paper proposes an out-of-distribution (OOD) detection method, where the features of in-distribution (ID) samples and those of the OOD samples are decorrelated in a class-wise manner by using HSIC.  A theoretical guarantee for decorrelation is provided.  The propposed method, Conditional-i, is evaluated on OOD detection benchmarks in image classification and NLP, and shows SOTA results.

Reviewers had many major concerns, e.g., novelty from HOOD, missing baselines with OOD exposure, hyperparameter tuning, and insufficient theory to support the proposed method, which the authors addressed well, and most of the reviewers have been convinced.

The paper is well-written and the mathematical notation seems fine for machine learners.  However, the relation between Theorem 1 and Assumption 1 is weird.  Any theorem must always hold (otherwise it's not a theorem), and the sentence "Theorem 1 holds if Assumption 1 holds" doesn't make sense.  If the claim in Theorem 1 requires Assumption 1, Assumption 1 should appear first, and Theorem 1 should state that "under Assumption1, it holds that ...".

**Award:**

No

---

### Decision · Program_Chairs · 2022-09-14

Accept